# Rate and oscillatory switching dynamics of a multilayer visual microcircuit model

Gerald Hahn[1]*, Arvind Kumar[2], Helmut Schmidt[3], Thomas R Knösche[3,4], Gustavo Deco[1,5,6,7]

[1]Center for Brain and Cognition, Computational Neuroscience Group, Department of Information and Communication Technologies, Universitat Pompeu Fabra, Barcelona, Spain; [2]Computational Science and Technology, School of Electrical Engineering and Computer Science, KTH Royal Institute of Technology, Stockholm, Sweden; [3]Brain Networks Group, Max Planck Institute for Human Cognitive and Brain Sciences, Leipzig, Germany; [4]Institute of Biomedical Engineering and Informatics, Department of Computer Science and Automation, Technische Universität Ilmenau, Ilmenau, Germany; [5]Institució Catalana de la Recerca i Estudis Avançats, Barcelona, Spain; [6]Department of Neuropsychology, Max Planck Institute for Human Cognitive and Brain Sciences, Leipzig, Germany; [7]School of Psychological Sciences, Turner Institute for Brain and Mental Health, Monash University, Melbourne, Australia

*For correspondence:
gerald.a.hahn@gmail.com

Competing interest: The authors declare that no competing interests exist.

**Abstract** The neocortex is organized around layered microcircuits consisting of a variety of excitatory and inhibitory neuronal types which perform rate- and oscillation-based computations. Using modeling, we show that both superficial and deep layers of the primary mouse visual cortex implement two ultrasensitive and bistable switches built on mutual inhibitory connectivity motives between somatostatin, parvalbumin, and vasoactive intestinal polypeptide cells. The switches toggle pyramidal neurons between high and low firing rate states that are synchronized across layers through translaminar connectivity. Moreover, inhibited and disinhibited states are characterized by low- and high-frequency oscillations, respectively, with layer-specific differences in frequency and power which show asymmetric changes during state transitions. These findings are consistent with a number of experimental observations and embed firing rate together with oscillatory changes within a switch interpretation of the microcircuit.

## Editor's evaluation

The microcircuit has a canonical composition and the interactions among distinct classes of excitatory and GABAergic neurons are fundamental to our understanding of sensory processing and neuronal synchronization. The authors investigate emerging dynamics in laminar models of the visual cortex, consisting of distinct GABAergic cell types, with a connectivity model based on the latest anatomical findings. The authors identify bistable circuit switches emerging from the interactions between different cell types and these are characterized by inhibited and disinhibited states accompanied by low- and high-frequency oscillations, respectively. These findings suggest a canonical, non-linear circuit motif that can explain multiple experimental observations and adds significantly to our understanding of microcircuit dynamics.

## Introduction

The neocortex is a recurrent network of morphologically diverse inhibitory interneurons and excitatory pyramidal neurons (PYR) (*Gouwens et al., 2019*; *Huang and Paul, 2019*; *Ascoli et al., 2008*;

*Markram et al., 2004*). The majority of interneurons can be assigned to biochemically defined classes, such as parvalbumin (PV), somatostatin (SST), and vasoactive intestinal polypeptide (VIP) positive cells (*Tremblay et al., 2016*). These neurons are distributed across layers and connected according to an intricate circuit diagram with intra- and interlaminar connections (*Pfeffer et al., 2013*; *Jiang et al., 2015*; *Kätzel et al., 2011*; *Harris and Shepherd, 2015*; *Cardin, 2018*). The discovery of regularities within the connectivity pattern of excitatory and inhibitory neurons prompted researchers to propose the existence of canonical microcircuits (*Douglas and Martin, 2004*; *Beul and Hilgetag, 2014*), which implement elementary computations that are repeated across the brain (*Miller, 2016*).

To identify such computations, research has focused on a better description of the functional role of individual neuron types by selective optogenetic activation and silencing of specific cell types (*Tremblay et al., 2016*; *Kepecs and Fishell, 2014*; *Fishell and Kepecs, 2020*; *Maffei, 2017*). These studies have not only highlighted an essential role of inhibitory neurons to balance excitation, but also recognized disinhibitory subcircuits, which release pyramidal neurons from strong inhibition (*Fishell and Kepecs, 2020*; *Letzkus et al., 2015*; *Pi et al., 2013*; *Walker et al., 2016*; *Jackson et al., 2016*; *Fu et al., 2014*). Moreover, neuronal oscillations in different frequency bands have been attributed to the activity of different interneuron types (*Cardin et al., 2009*; *Sohal et al., 2009*; *Chen, 2017*). While the computational properties of simplified circuits with multiple interneuron types have been investigated theoretically (*Lee et al., 2017*; *Hertäg and Sprekeler, 2019*; *Garcia Del Molino et al., 2017*; *Lee, 2018*), in the context of vision, locomotion, prediction errors, and whole-brain models (*Lee and Mihalas, 2017*; *Dipoppa et al., 2018*; *Hertäg and Sprekeler, 2020*; *Bensaid et al., 2019*), the dynamics of more complex networks comprising multiple layers with translaminar connectivity remain unexplored. Moreover, even though models have examined the emergence of oscillations in local (*Lee, 2018*; *Domhof and Tiesinga, 2021*; *Veit et al., 2017*) and whole-brain neuronal networks (*Bensaid et al., 2019*) composed of canonical microcircuits, it is unclear how firing rate descriptions of microcircuit function relate to oscillatory behavior of cortical networks, which can differ across cortical layers (*Bastos et al., 2018*; *Adesnik, 2018*; *van Kerkoerle et al., 2014*). This is crucial to interpret meso- and macroscopic signals from local field potential (LFP), electroencephalography (EEG) and magnetencephalography (MEG) recordings with respect to circuit function, where access to firing rate information is not possible.

To address these questions, we take a computational approach and isolate the role of different neurons in rate- and oscillation-based functioning of the layered microcircuit of the primary mouse visual cortex, for which the most comprehensive connectivity diagram to date is available (*Jiang et al., 2015*). Modeling permitted to go beyond of what is possible with available experimental tools and we not only characterized the effect of selective activation/suppression of different neuron types, but also perturbed specific connections and test their impact on microcircuit dynamics and response properties.

We found that the superficial and deep layers in the visual microcircuit can operate in two different states, triggered by the external activation of specific interneuron types and each with different excitation–inhibition balance: An inhibition dominated state controlled by SST neurons (SST state) and a disinhibited state that is attained by activation of PV and/or VIP neurons (PV/VIP states). By perturbing connections of different types of interneurons, we confirmed that disparities in recurrent connections within these inhibitory cell classes play a crucial role for the different EI balance in the two states. Two mutual inhibitory motifs that include SST, PV, and VIP cells serve as ultrasensitive or bistable switches with different sensitivity, which can toggle the microcircuit between the two states. Such a state change in one layer can propagate through translaminar connections to the other layer. Notably, we also found that in the inhibited regime slow beta-band oscillations were more prevalent especially in the deep layer, whereas in the disinhibited state fast gamma oscillations emerged predominately in the superficial layer, similar to experimental observations (*Bastos et al., 2018*; *van Kerkoerle et al., 2014*). We also provide a mechanistic explanation to other empirical findings such as asymmetric changes in oscillation power and frequency during state transitions as seen during the presentation of visual stimuli with increasing size (*Chen, 2017*; *Veit et al., 2017*). Thus, our results provide a comprehensive description of state-dependent effects of different inhibitory interneuron types with testable predictions and link rate- and oscillation-based accounts of microcircuit functioning.

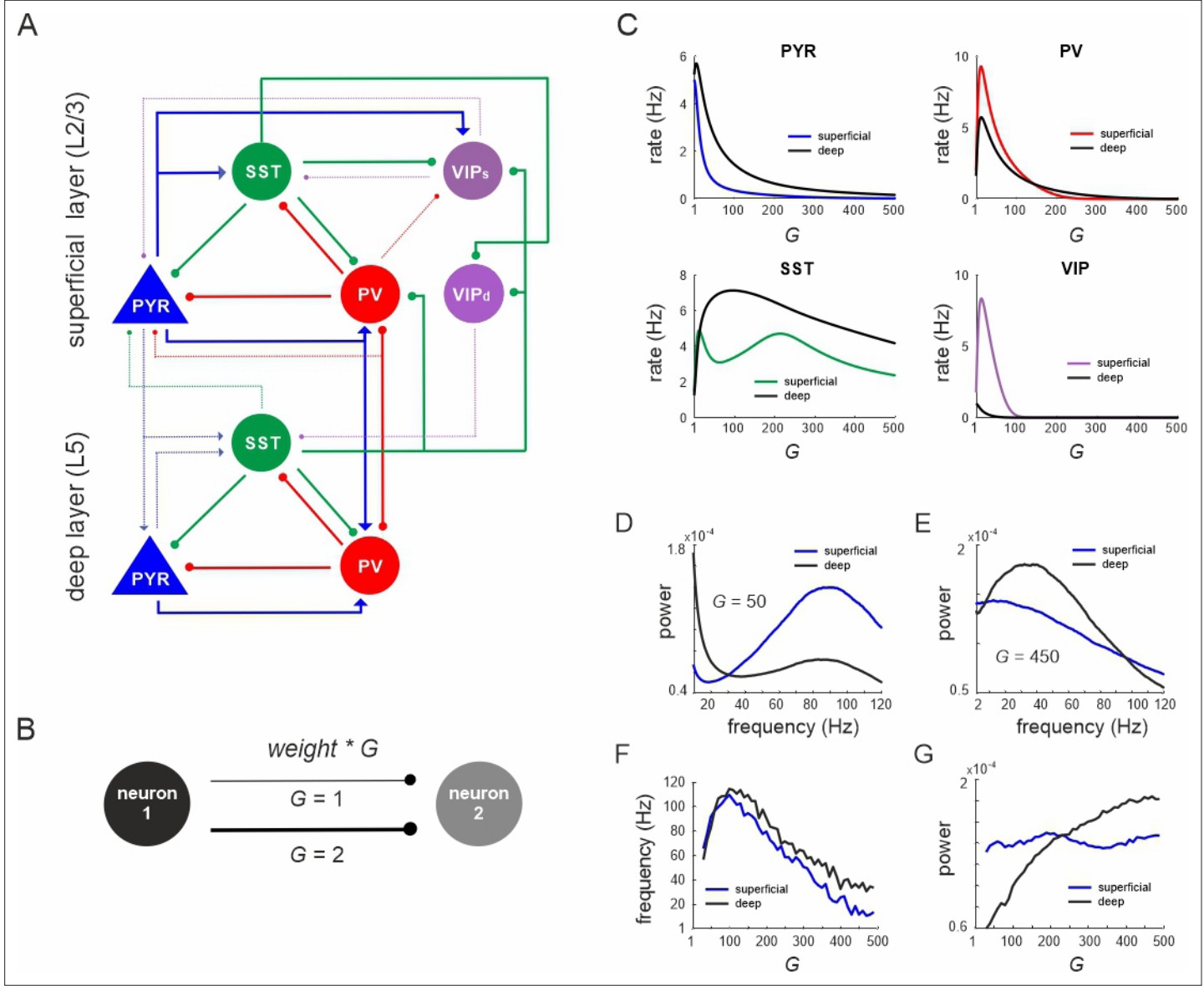

**Figure 1.** Network anatomy and spontaneous activity. (**A**) Layout of the local network with a superficial layer that includes four different cell types in layer 2/3 of the mouse visual cortex and three cell types in a deep layer representing L5. Even though residing in the superficial layer, the VIP$_d$ cell type was functionally associated with the deep layer, as it mainly innervates L5. The connectivity strength ($w$) is represented by the thickness of the lines. Solid lines: $w > 0.1$, dashed lines: intermediate weights: $0.04 > w < 0.1$, weak weights ($w < 0.04$) are not shown. (**B**) Schematic showing the scaling of a connection by a coupling parameter $G$. (**C**) Mean spontaneous rate for all cell types in superficial and deep layers as a function of the coupling parameter $G$. (**D, E**) Example power spectra of local field potential (LFP) in superficial and deep layers for two different values of $G$. (**F, G**) Frequency and power of oscillatory peaks in LFP spectra as a function of $G$ for both layers.

## Results

In this study, we investigate the computational properties of a detailed microcircuit of the mouse visual cortex (*Jiang et al., 2015*; *Figure 1A*). This network consists of two different layers (superficial and deep), representing L2/3 and L5 of the primary visual cortex and each containing four different cell types that are connected within and between layers: excitatory pyramidal cells (PYR), and three different classes of inhibitory cells (PV, SST, and VIP). The connectivity was corrected for the different prevalence of each cell type (*Jiang et al., 2015*) and scaled with a global parameter ($G$) to approximate effective coupling changes (*Deco et al., 2013*; *Jobst et al., 2017*). This scaling was necessary, because the connectivity weights were empirically determined for individual cell pairs, whereas the effective interactions are mediated through entire cell populations for which the absolute cell count

is not known. The variable $G$ controls for the effective connectivity between cell types, such that a larger value of $G$ implies a larger number of cells in the network. Thus, the connectivity strength between the modeled cell classes in our model also approximates the population size (*Figure 1B*). The population dynamics of each neuron type was given by a firing rate model (see Methods). The different excitatory–inhibitory feedback loops endowed the model with resonance properties, which was accompanied by oscillations in the presence of externally applied noise. This allowed us to study both firing rate and oscillatory behavior as measured by variations in power and frequency of the LFP, approximated by the rate of the pyramidal cell populations. We first examined spontaneous interactions across all neurons and then drove specific neuron classes, simulating input from remote cortical and subcortical sources.

### Spontaneous activity

First, we systematically scaled the microcircuit connectivity by $G$, and measured the steady-state firing rates of all neurons without any external input. A sharp increase in pyramidal, PV and VIP cell activity with $G$ was followed by a rapid decrease of mean rates in both layers (*Figure 1C*). SST neurons also behaved similar to PV and VIP cells, but both rise and decay in their activity were much slower. The average firing rates of pyramidal and SST neurons were higher in the deeper layer, in accordance with experimental results in mice (*Sakata and Harris, 2009*; *Senzai et al., 2019*). Power spectral analysis of the LFP showed a clear peak, whose frequency and power varied with the coupling parameter in a layer-specific manner (*Figure 1D, E*). Generally, frequencies first increased within the high gamma range from ~60 Hz (for small $G$) to ~110 Hz (for $G = 100$) across both layers (*Figure 1F*). For $G > 100$, the dominant LFP frequency steadily decreased to a low gamma (~40 Hz, superficial layer) or low beta range (~15 Hz, deep layer) for $G = 500$. Interestingly, the frequency consistently remained higher in deep layers, consistent with recent experimental findings in mouse V1 (*Adesnik, 2018*). Moreover, high gamma frequencies ($G < 250$) were stronger in superficial layers, whereas the power of slower oscillations ($G > 250$) was higher in deep layers (*Figure 1G*), in congruence with empirical mouse V1 findings (*Senzai et al., 2019*).

### Origin of different firing rates and oscillations in deep and superficial layers

Next, we investigated the anatomical origin of firing rate and oscillation differences across layers by modifying specific connections. We targeted three connections, which show pronounced asymmetry across layers: PYR$^{sup}$ → PYR$^{deep}$ connection, translaminar projections of SST cells, and recurrent inhibition among PV neurons (PV–PV connections) (*Figure 2A*, *Supplementary file 1d*). The removal of the PYR$^{sup}$ → PYR$^{deep}$ connection strongly reduced the firing rate differences across the layers (*Figure 2B*). Removal of the translaminar SST connection, which only projects from deep to superficial layer had a smaller impact on the firing rate difference. Moreover, the disinhibitory PV–PV connections are considerably stronger in deep layer (*Supplementary file 1d*). When we changed the PV–PV connections such that their strength was the same in the deep and superficial layers, the firing rate difference between the two layers was also reduced. Modifying all three connections simultaneously almost completely abolished the rate inequality between layers. Likewise, differences in oscillation power in different frequency bands across layers were suppressed (*Figure 2C, D*, compare with *Figure 1D, E*). Thus, our model suggests that stronger excitation and disinhibition in the deep layer together with more inhibition in the superficial layer underlie the experimentally observed firing rate and oscillation power differences between deep and superficial layers.

### Effect of silencing specific inhibitory cells

Next, we further investigated the role of different interneurons in shaping the frequency and power of oscillations in the low and high activity state, respectively. To this end, we silenced individual inhibitory cell types in both layers by removing all their connections and examined the effect on oscillation frequency and power of the LFP (*Figure 2E*). Knocking out PV cells was accompanied by a slow oscillation (~30 Hz) in both layers with a frequency that remained approximately stable with $G$. By contrast, when SST cells were removed the network oscillated at high frequency (~110 Hz) across all tested values of $G$ (*Figure 2F*, top, *Figure 2—figure supplement 2A*, left). Note that this increase in the oscillation frequency was not due to disinhibition from SST silencing, because PYR firing rate

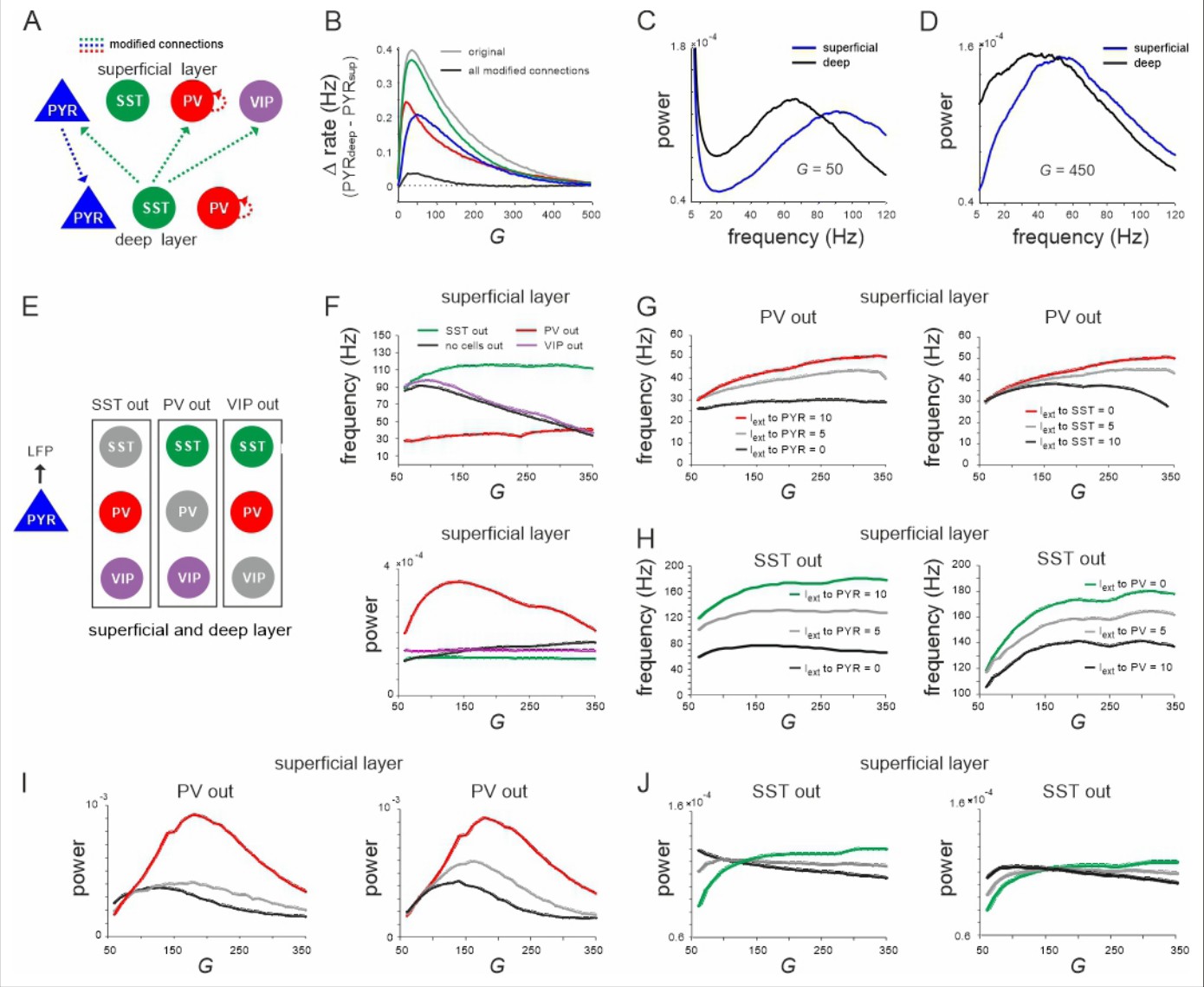

**Figure 2.** Effect of connectivity lesions on rate and spectral properties across layers. (**A**) Schematic two-layer network. The blue and green translaminar connections were removed, while the weights of the red connections were set to the same value of the deep recurrent parvalbumin (PV) connection. (**B**) Effect of connectivity modification on rate difference between superficial and deep layer as a function of $G$. (**C, D**) Power spectra of both layers for two values of $G$ after all connectivity modifications were applied. (**E**) Diagram depicting three different cell lesion simulations. Gray circles: connections of this cell type to all other cells were set to zero. (**F**) Peak local field potential (LFP) frequency (top) and power (bottom) as a function of $G$ for different cell lesions in the superficial layer. (**G**) Peak frequency in the superficial layer as a function of $G$ after PV cell inactivation and different levels of input to PYR (left) or somatostatin (SST) cells (right). (**H**) Superficial layer peak frequency after SST lesion as a function of $G$ and varying input to PYR or PV cells. (**I, J**) Same as in (**G, H**) for oscillatory peak power in the superficial layer.

The online version of this article includes the following figure supplement(s) for figure 2:

**Figure supplement 1.** Comparison of the evolution of somatostatin (SST) and parvalbumin (PV) rates with increasing $G$.

**Figure supplement 2.** Effect of silencing specific cell types on rate and spectral properties in the deep layer.

decreased in both knockout cases with $G$ (*Figure 2—figure supplement 1*), while the frequency remained high. In accordance with this hypothesis, for small values of $G$, when PV rate was high, the oscillation frequency in the intact network approached the frequency seen in the SST knockout case. As $G$ was increased and SST activity surpassed PV firing rates, the oscillation frequency decreased and converged to values seen in the PV knockout scenario. Importantly, oscillatory power was overall higher in the PV knockout case with a low frequency and lower in the SST silenced network with faster oscillations across all tested values of $G$ (*Figure 2F*, bottom, *Figure 2—figure supplement 2A*, right).

In contrast to PV and SST cells, VIP cell silencing only slightly increased the oscillation frequency, but did not influence the relative decrease in frequency as a function of *G*. However, when we manipulated frequency and power in the knockout networks, we found that they changed symmetrically. In both PV and SST silenced circuits, driving PYR cells or the remaining inhibitory cells jointly increased or decreased power and frequency in superficial and deep layer, except for the deep layer after SST silencing (*Figure 2G–J*, *Figure 2—figure supplement 2B, C*). Thus, the relative dominance of PV and SST cell activity is an important factor that determines the oscillation frequency and power, which are inversely related in the full model (see below), but positively correlated in the partly silenced network.

## Two different states and state switching dynamics of the microcircuit

Thus far, we changed the relative prevalence of given interneuron types by scaling the connectivity matrix or silencing individual cell types or connections. Visual inspection of the microcircuit revealed two prominent mutual inhibition motifs. SST cells exhibit reciprocal inhibitory connections with PV cells and also VIP cells in each layer, which brings these cell pairs in competition with each other (*Figure 1A*). We hypothesized that driving one inhibitory cell type will functionally silence competing cells and toggle the circuit between different inhibitory states, which may differ in terms of PYR firing rate and their oscillation profile. To verify this hypothesis, we first tested whether input to VIP or PV cells can suppress SST activity. To this end, we enhanced the SST activity by injecting additional input to SST cells in both layers ($I_{ext}$ = 5 Hz). Next, we stimulated either VIP or PV cells in both layers simultaneously, mimicking feedforward input from layer 4 to PV cells or feedback input from upstream areas to VIP cells, which may target superficial and deep layers simultaneously (*van Kerkoerle et al., 2014*; *Pluta et al., 2019*). Responses were measured from PYR and SST cells for different values of *G* (*Figure 3A, B*, see *Figure 3—figure supplement 1A, B* for responses of other cell types).

When VIP cells were driven (*Figure 3—figure supplement 1A*), we found two main effects. First, VIP connections to SST cells suppressed SST rates, whereas PYR and PV cells initially increased their firing across both layers for sufficiently high *G* values (*G* > 150), because they were released from SST inhibition (*Figure 3A* and *Figure 3—figure supplement 1A*, *G* < 25). However, as VIP input increased, PYR and PV cells were gradually suppressed again in the superficial layer, while their firing rate was only slightly affected in the deep layer. This was caused by the inhibitory connection from VIP to PYR cells in the superficial layer and indeed its removal resulted in a response similar to the deep layer (*Figure 1A*, *Figure 3—figure supplement 2*). The presence and strength of disinhibition in the superficial layer depended on the baseline activity of SST cells, with higher SST rates, either due to high *G* values or caused by external drive, enhancing the PYR cell disinhibitory peak (*Figure 3—figure supplement 1B*). By contrast, when SST rates were too low, VIP input failed to further suppress SST activity and thus no disinhibition of PYR cells was seen.

Second, as *G* increased, the responses of SST cells became more switch like with sigmoidal curves in both layers, a hallmark of an ultrasensitive switch in many biological systems (*Ferrell and Ha, 2014*; *Figure 3A*). A similar sigmoidal decrease in SST firing rates with initial PYR disinhibition followed by inhibition was observed in both layers when only PV cells were driven (*Figure 3B*, *Figure 3—figure supplement 1C*). However, when PV cells were stimulated, switch like ultrasensitive responses occurred at larger *G* values as compared to VIP input, especially in the deep layer. By contrast, VIP cell-induced inhibition on PYR cells was weaker than PYR suppression mediated by PV cells (*Figure 3A, B*, *Figure 3—figure supplement 1D*).

In some biological systems ultrasensitivity with sigmoidal response curves is accompanied by bistable behavior (*Ferrell, 2002*), characterized by state transitions that are not reversed when the input is withdrawn. A telltale sign of bistability is the presence of hysteresis, that is the response curves change as a function of the direction in which the state change was triggered. To test for bistability in the microcircuit, we first applied an increasing current to VIP or PV cells in both layers simultaneously, followed by current in the decreasing direction for different values of the coupling parameter. We found that hysteresis appeared at sufficiently high *G* values, as visible by the appearance of noncongruent response curves for the up and down direction in all cell types of both layers for VIP (*Figure 3C*, *Figure 3—figure supplement 3A*) and PV input (*Figure 3D*, *Figure 3—figure supplement 3B*). Hysteresis in the deep layer for PV input required very strong input, even though a small hysteresis effect was observed for smaller *G* that was transmitted from superficial layers via translaminar connections (*Figure 3—figure supplement 3C*).

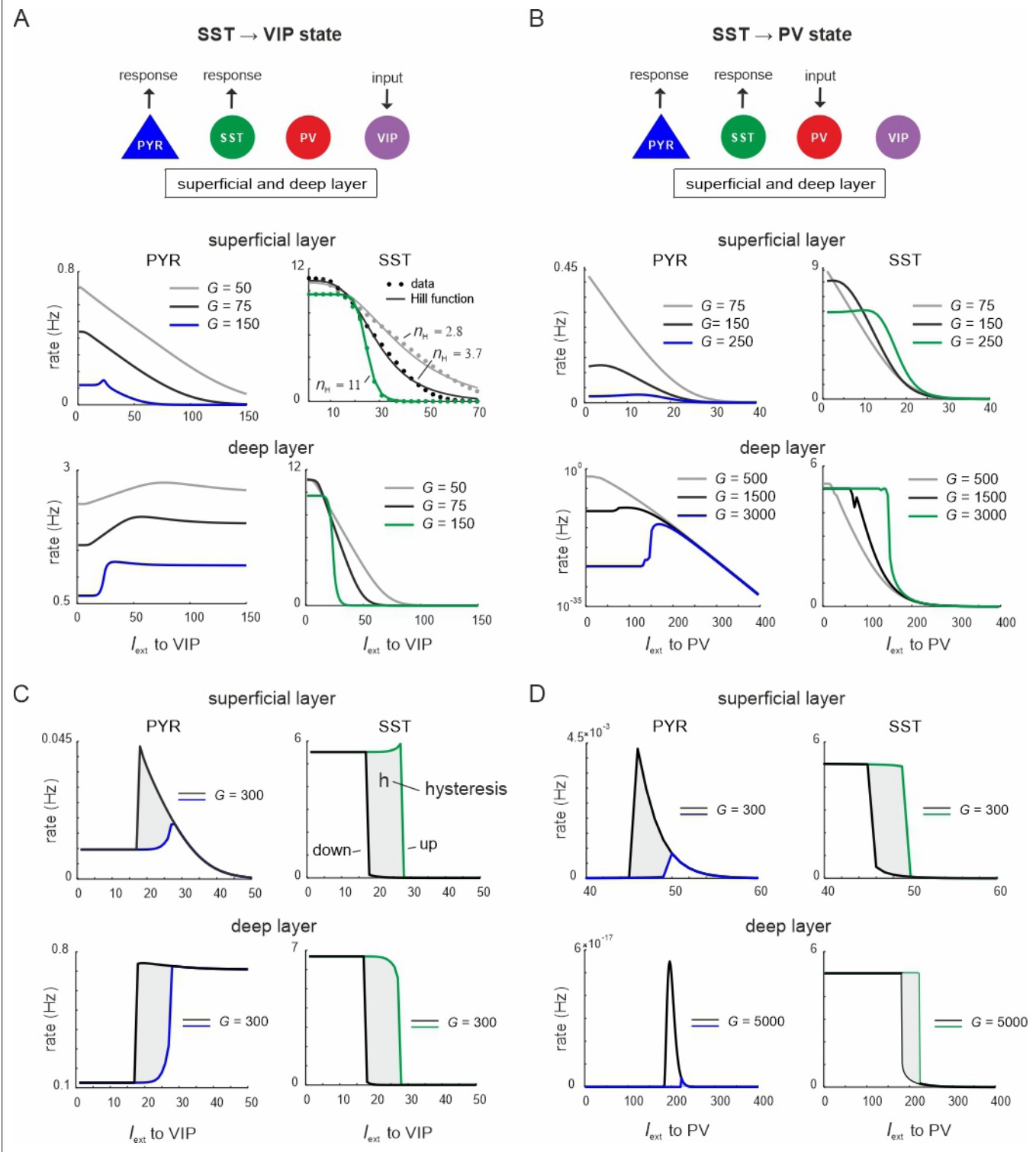

**Figure 3.** Ultrasensitivity and hysteresis in the visual microcircuit. (**A**) Response of PYR and somatostatin (SST) cells in superficial and deep layers after input to superficial and deep vasoactive intestinal polypeptide (VIP) cells (top), mimicking cortical feedback. SST cells were driven with a constant input $I_{ext}$ = 5 5Hz to enhance SST activity. Responses are shown for three different values of $G$ (bottom). SST response curves were fitted with the Hill function, yielding a different Hill coefficient ($n_H$) for each curve. (**B**) Same as in (**A**) for simultaneous input to superficial and deep parvalbumin (PV) cells, simulating feedforward inhibition. (**C**) PYR and SST cell response to increasing (up branch, in blue/green) and decreasing input (down branch, in black) to VIP cells for an exemplary value of $G$ and both layers. Note that $G$ is higher than in (**A**). The network displays hysteresis in each layer (shaded region $h$). (**D**) Same as in (**C**) for simultaneous input to PV cells in both layers.

*Figure 3 continued on next page*

*Figure 3 continued*

The online version of this article includes the following figure supplement(s) for figure 3:

**Figure supplement 1.** Ultrasensitivity in parvalbumin (PV) and vasoactive intestinal polypeptide (VIP) cells.

**Figure supplement 2.** Vasoactive intestinal polypeptide (VIP) cells only inhibit superficial PYR cells.

**Figure supplement 3.** Hysteresis in parvalbumin (PV) and vasoactive intestinal polypeptide (VIP) cells.

**Figure supplement 4.** Mutual inhibitory connectivity weights control ultrasensitivity and hysteresis.

Inhibition-based ultrasensitivity and hysteresis commonly require strong inhibitory interactions between the components of the system (**Ferrell, 2002**). Therefore, we hypothesized that the enhanced mutual inhibitory connections between SST <> VIP and SST <> PV cells due to the scaling by $G$ underlie the sigmoidal and hysteretic response curves. Indeed, selectively increasing these weights was sufficient for ultrasensitivity and hysteresis to appear in the microcircuit (**Figure 3—figure supplement 4**).

Next, to quantify the switching behavior of the microcircuit we fitted a Hill function to the SST firing rate response curve (see Methods) and estimated the Hill coefficient ($n_H$, **Figure 3A**), a measure for ultrasensitivity. We also estimated the area between the up and down branches of the response curves ($h$) to quantify hysteresis. We found that both $n_H$ and $h$ increased monotonically with $G$ when either VIP cells (**Figure 4A**) or PV cells (**Figure 4B**) were stimulated. Within a limited range of $G$, the Hill coefficient ($n_H$) increased monotonically for the transition from SST to VIP or SST to PV states in both

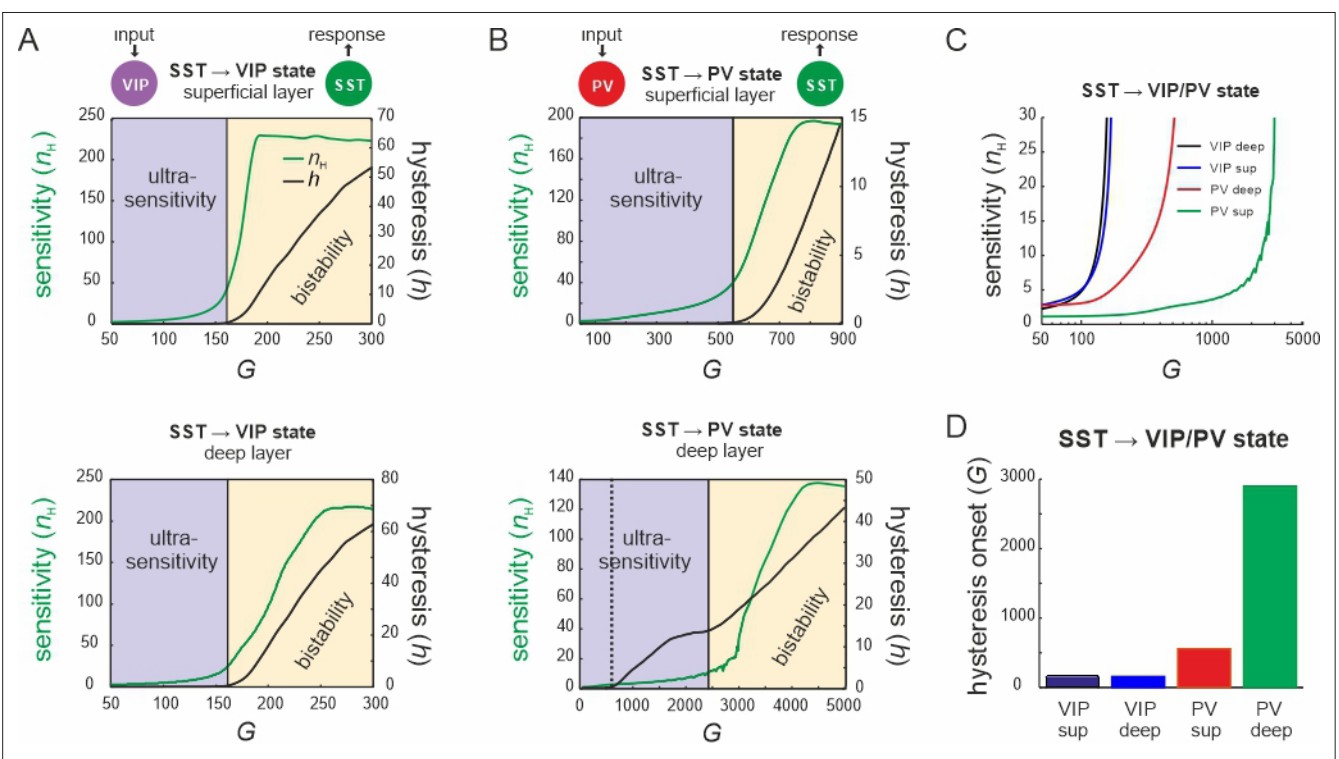

**Figure 4.** Bifurcation diagrams of the microcircuit switches. (**A**) Bifurcation diagram of the superficial (top) and deep layer (bottom) depicting sensitivity, as measured by the Hill coefficient ($n_H$), and hysteresis area ($h$) of somatostatin (SST) cell responses to vasoactive intestinal polypeptide (VIP) cell input as a function of $G$. The shaded areas show a purely ultrasensitive and a bistable region, where both ultrasensitivity and hysteresis are present. (**B**) Same as in (**A**) for input to parvalbumin (PV) cells. Bottom dotted line: onset of hysteresis in the superficial layer (top) is also seen in the deep layer (bottom). (**C**) Sensitivity of SST responses to VIP or PV input in superficial and deep layers as a function of $G$. (**D**) $G$ values of hysteresis onset in SST responses in both layers after input to VIP and PV cells.

The online version of this article includes the following figure supplement(s) for figure 4:

**Figure supplement 1.** Vasoactive intestinal polypeptide (VIP) switch properties in the presence of varying levels of baseline input ($I_{base}$).

**Figure supplement 2.** Vasoactive intestinal polypeptide (VIP) switch properties in the presence of varying levels of noise and different transfer function parameters.

layers, while hysteresis was absent ($h = 0$). Hysteresis occurred at different values of $G$ depending on the type of switch and layer, and caused an abrupt increase in $n_H$, which stabilized with large values of $G$ and gave rise to a virtually binary state transition. A closer study of the bifurcation diagrams showed marked differences across switches and layers. As we increased $G$, the sensitivity increased rapidly for the VIP to SST switch in both layers. By contrast, a similar increase of sensitivity for the PV to SST switch occurred at higher values of $G$ in the superficial layer and even higher $G$ values in the deep layer (*Figure 4C*), as compared to the VIP to SST switch. Likewise, hysteresis onset increased from the VIP switch to the PV switch in the superficial and deep layer (*Figure 4D*). Note that the PV switch in the deep layer showed an early increase of $h$ due to a propagated hysteresis effect from the superficial layer ($G \sim 500$, *Figure 3—figure supplement 3C*) and showed its own hysteresis increase later.

Next, we tested whether the results on the VIP switch hold when external baseline input was added to both layers (*Figure 4—figure supplement 1A*). The same disinhibition effect was present in PYR cells and was enhanced with stronger baseline activity (*Figure 4—figure supplement 1B*). We also found the same sigmoidal SST suppression curve, which was shifted to higher values and displayed a larger hysteresis area with more baseline input (*Figure 4—figure supplement 1C, D*). However, the slope of the curve was preserved and the point of transition between ultrasensitivity and hysteresis was not altered.

## Robustness of the switching dynamics

In the above, we characterized the dynamics of the microcircuit switch in noise free conditions (*Figure 3*). To test the robustness of our results, we added Gaussian noise to the different cell populations (see Methods). We found that the addition of noise does not change the results (*Figure 4—figure supplement 2A*).

Next, we tested to what extent the specific choice of connectivity affects the switching dynamics. Experimental data show that the connection strength between neurons is not fixed, but varies within a limited range (*Jiang et al., 2015*). Thus, we jittered the connection weights in the model in accordance with the experimental literature (see Methods). The jitter in the connection weights indeed changed the slope of the SST response curves (*Figure 4—figure supplement 2B*), which resulted in a large variety of Hill coefficient values (*Figure 4—figure supplement 2C*).

Finally, we examined whether intrinsic properties of the neuron populations affect the switching dynamics. Individual neurons have different intrinsic properties, which change how input is translated into firing rates (*Zerlaut et al., 2019*). Thus, we also varied the two parameters ($a$: slope and $b$: threshold) of the model transfer function. We found that both parameters have different effects on the switching properties (*Figure 4—figure supplement 2D, E*). While the parameter $a$ only shifted the SST response curves to higher values (*Figure 4—figure supplement 2D*), the parameter $b$ increased the slope (*Figure 4—figure supplement 2E*). Thus, our key results are robust to noise in the input, connectivity, and neuron properties.

Taken together these findings suggest that the microcircuit can operate in two different states: An inhibited state, where SST neurons dominate and a disinhibited state with prevailing PV and/or VIP activity. Two mutually inhibitory circuit motifs provide two switches with different sensitivity, which toggle the network between both states, characterized by different PYR rates, while maintaining sufficient inhibition to putatively prevent runaway excitation.

## Oscillations and switching dynamics

Next, we studied the outcome of the switching dynamics on oscillation frequency and power of the LFP. Both input to VIP and PV cells strongly increased the frequency of the dominant oscillation in superficial and deep layers, whereas the power of the oscillation peak generally decreased (*Figure 5A–D*). This is expected from a transition from an SST dominated state with low frequency to a high oscillation state in which the frequency is imposed by dominating PV activity (see above). Note that frequency and power show sigmoidal jumps similar to the rate transitions above (*Figure 2I, J*, VIP input).

These results were obtained using fixed time constants ($\tau$) that were specified for each cell class according to the literature, which is based on the notion that fast and slow oscillations are generated using different interneuron populations with short- and long-time constants, respectively (*Hyafil et al., 2015*; *Mejias et al., 2016*). To examine whether the observed transition between slow and fast oscillations for the VIP switch in the superficial layer indeed depends on interneuron-specific time constants,

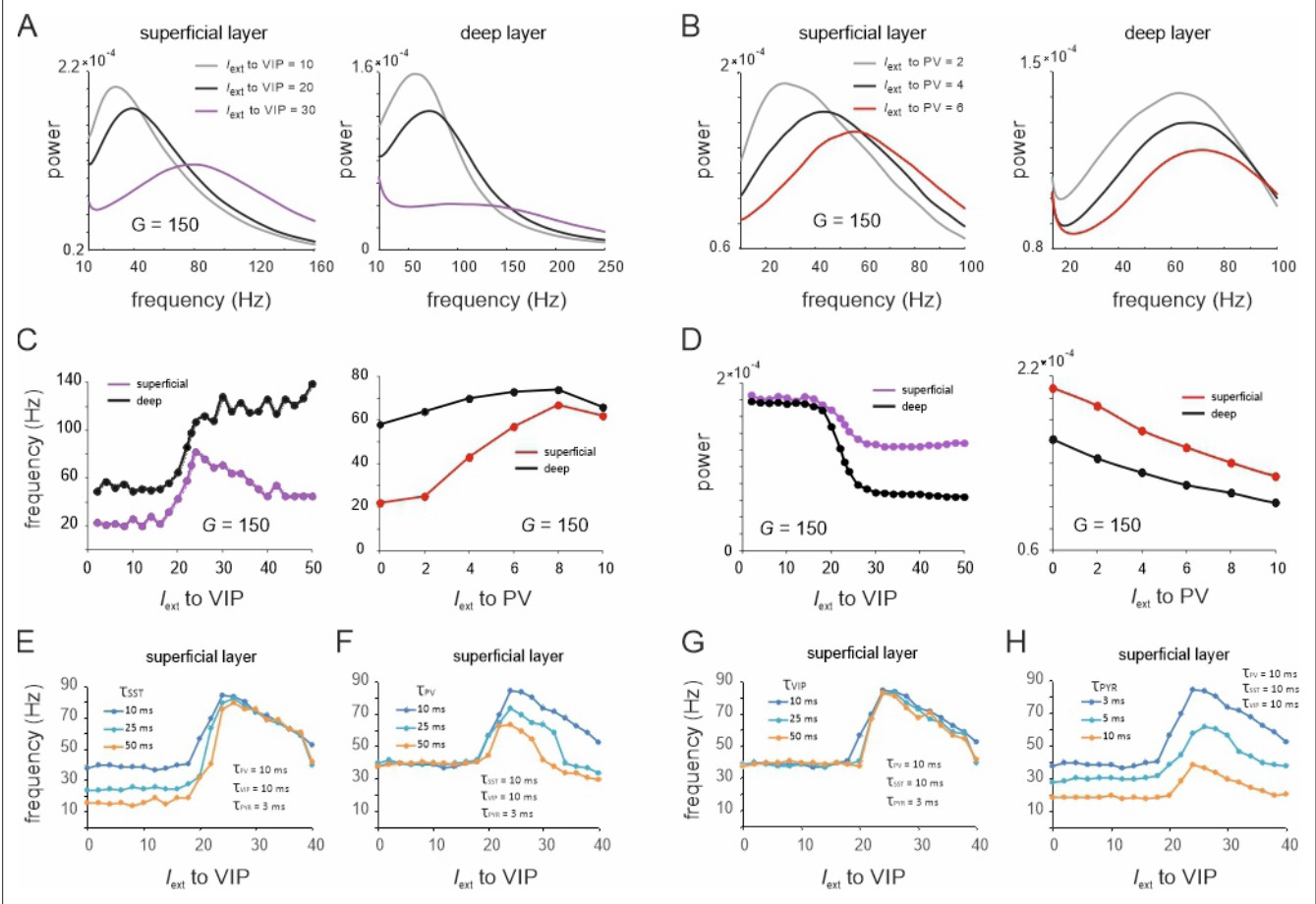

**Figure 5.** Spectral properties of the microcircuit switches. (**A**) Local field potential (LFP) power spectra of superficial and deep layers with three different inputs to vasoactive intestinal polypeptide (VIP) cells and a constant drive $I_{ext}$ = 5Hz to somatostatin (SST) cells. (**B**) Same as in (E) for input to parvalbumin (PV) cells. Peak frequency (**C**) and power (**D**) in superficial and deep layers as a function of VIP and PV input. (**E–H**) Peak frequency in the superficial layer as a function of VIP input and varying time constants of different cell types.

we systematically studied different values of $\tau$ across all neuron types. To this end, we fixed $\tau$ at 10 ms for three neuron types and systematically varied the fourth type. When the SST time constant was altered, we found that the frequency difference between slow and fast oscillation states increased due to a frequency reduction within the slow oscillation state, dominated by SST cells (**Figure 5E**). In contrast, the frequency within the high activity state, dominated by PV cells, remained unchanged. Remarkably, a marked transition between a slow and fast oscillation state was still visible, even when the time constant across all neurons was the same. Increasing $\tau$ of PV cells had the opposite effect since the frequency of the high activity state decreased, while there was no change in frequency of the low activity state (**Figure 5F**). A variation of the VIP cell time constant had no influence on the oscillation frequency in either state, demonstrating again its negligible influence on oscillatory dynamics (**Figure 5G**). Finally, increasing the PYR time constant reduced the frequencies in both the slow and fast oscillation states (**Figure 5H**). The disinhibition peak was reduced, but remained at approximately double the frequency of the SST dominated baseline frequency for all tested PYR time constants. In summary, these results demonstrate that the appearance of a frequency jump during switching is independent of the time constants, even though the extent of the frequency change is modulated by differences in $\tau$.

## Lateral inhibition switches the circuit to the SST state

Next, we studied switching dynamics in the microcircuit in the opposite direction, that is a transition from PV/VIP toward SST governed activity. To this end we applied input to SST cells in both layers (**Figure 6A**), mimicking the effect of lateral inhibition during surround suppression in the visual

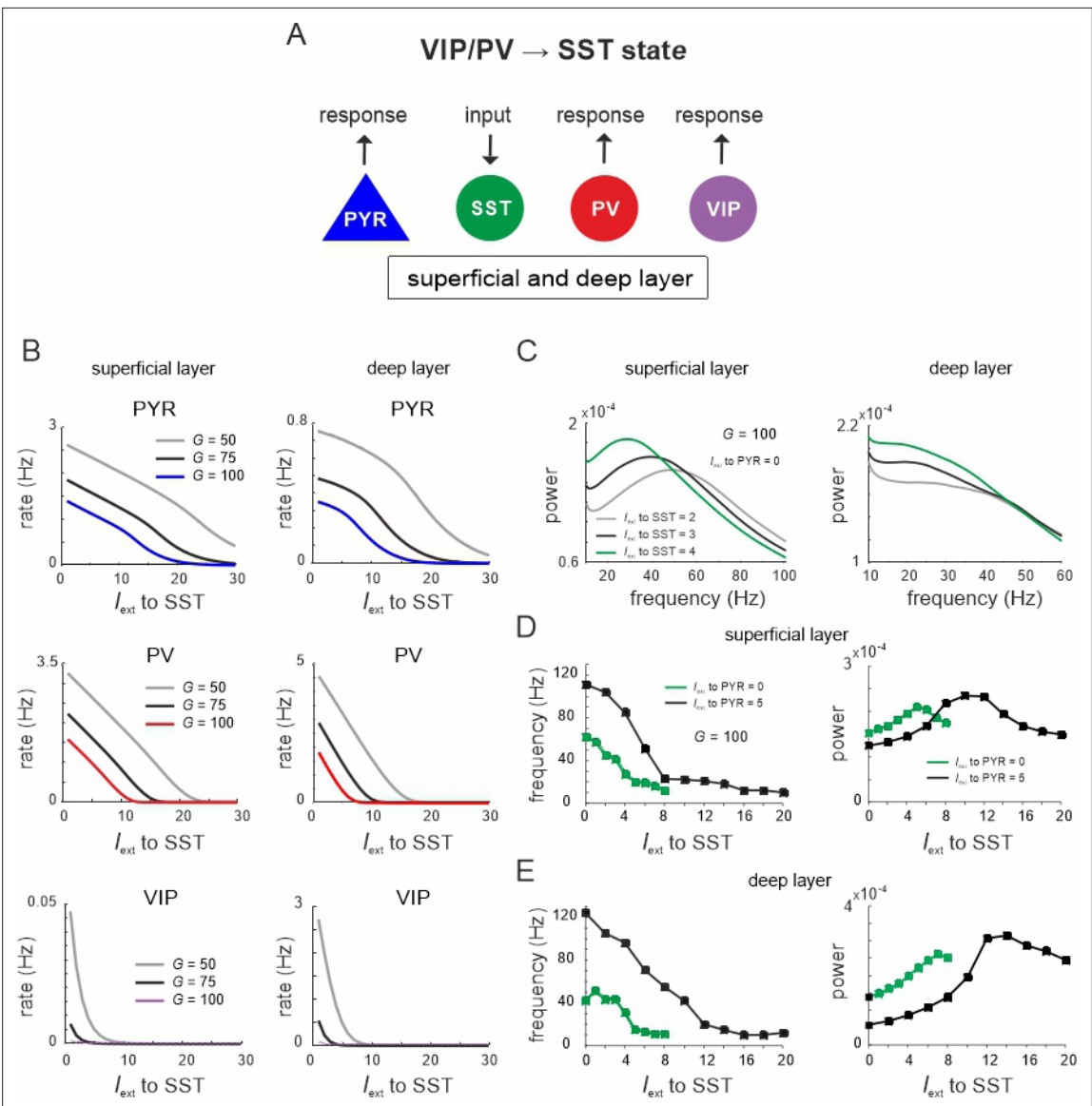

**Figure 6.** Lateral inhibition switches dynamics from vasoactive intestinal polypeptide (VIP)/parvalbumin (PV) to somatostatin (SST) dominated states. (**A**) Schematic showing input to SST cells in both layers and measuring response in all other cell types. (**B**) Response curves of PYR, PV, and VIP cells as a function of SST drive for superficial and deep layers, and three different values of G. (**C**) Power spectra of superficial and deep layers for different levels of SST input. (**D**) Peak frequency and power as a function of SST input for two levels of PYR drive. (**E**) Same as in (**D**) for the deep layer.

cortex, which was experimentally found to be mediated by horizontal pyramidal cell input from distant microcircuits within the same area to SST neurons (*Veit et al., 2017*; *Adesnik et al., 2012*). Driving SST cells resulted in a monotonic decay of activity of all other cell types in both layers for different values of G (*Figure 6B*), in line with several experimental studies (*Chen, 2017*; *Dipoppa et al., 2018*; *Adesnik et al., 2012*). Stimulation of SST cells also reduced the frequency of oscillations (*Figure 6C–E*). However, as we increased the input to SST cells the power of oscillations initially increased and subsequently decreased once PV cells were strongly suppressed. This finding replicates experimental results, in which stronger surround suppression is followed by a sudden transition from high frequency (gamma range) to lower frequency oscillations (high beta, low gamma range) and a concomitant increase in oscillatory power in mouse V1 (*Chen, 2017*; *Veit et al., 2017*).

### Input to PYR cells favors SST activity

Next, we measured the response of SST cells when PYR cells were stimulated in both layers (*Figure 7A–C*, *Figure 7—figure supplement 1A*). We studied the response of SST cells in three

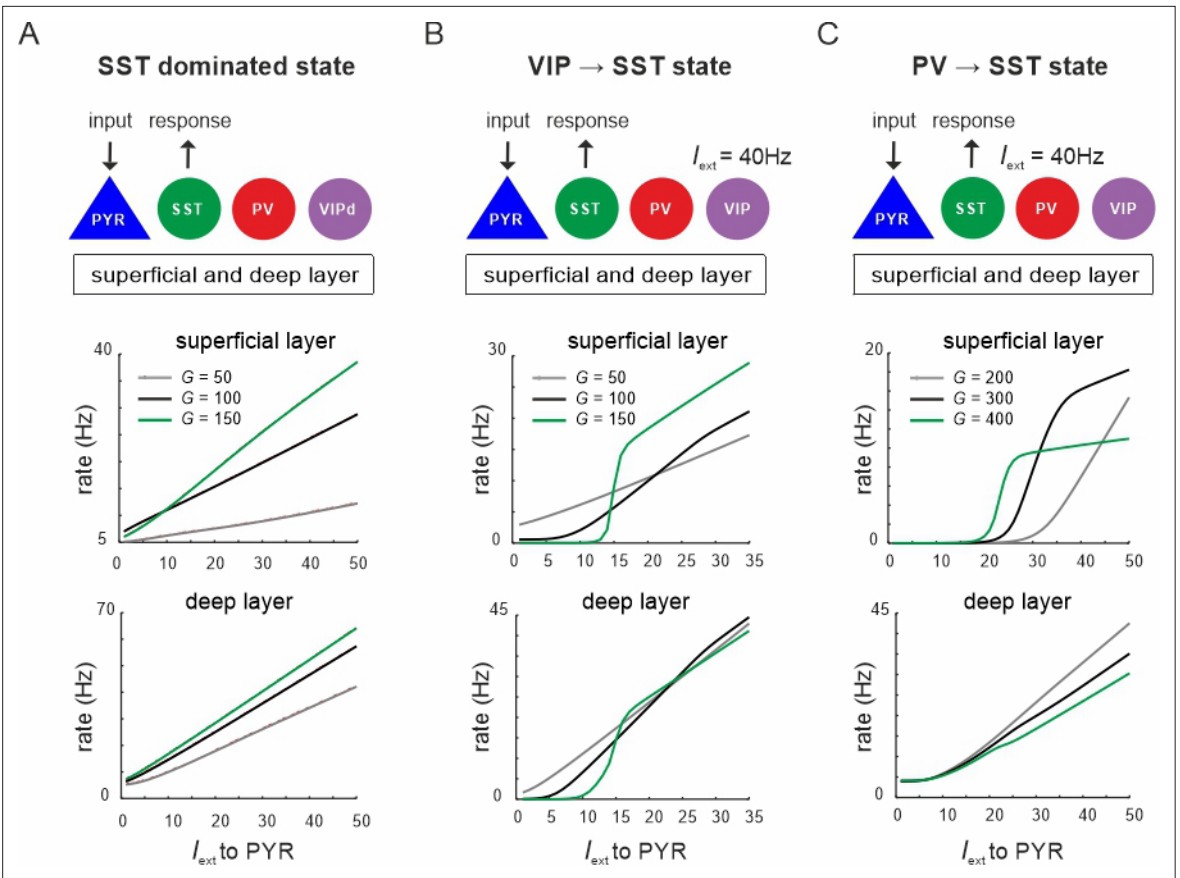

**Figure 7.** Input to PYR cells poises the microcircuit to somatostatin (SST) dominated dynamics. (**A**) Schematic of PYR cell input (top) and response of SST cells to increasing PYR drive in both layers for three values of G (top and bottom). (**B**) Same as in (**A**) with vasoactive intestinal polypeptide (VIP) cells being driven with a constant current $I_{ext}$ = 40Hz. (**C**) Same as in (A) with constant input $I_{ext}$ = 40Hz to parvalbumin (PV) cells.

The online version of this article includes the following figure supplement(s) for figure 7:

**Figure supplement 1.** Response of different cell types to PYR cell input during the somatostatin (SST) dominated state.

**Figure supplement 2.** Response of different cell types to PYR cell input during the vasoactive intestinal polypeptide (VIP) dominated state.

**Figure supplement 3.** Response of different cell types to PYR cell input during the parvalbumin (PV) dominated state.

**Figure supplement 4.** VIP switch properties with different recurrent connectivity strength of PYR cells.

different scenarios. In the first scenario, we stimulated PYR cells, while VIP and PV cells received no external input. In this case, PYR input strongly increased SST activity (***Figure 7A***) (and to a lesser extent PV rates, see ***Figure 7—figure supplement 1B***) in both layers, while VIP cells were suppressed (***Figure 7—figure supplement 1B***). Stimulation of PYR cells also increased the population oscillation frequency, whereas oscillation power decreased (***Figure 7—figure supplement 1C***). In a second scenario, we stimulated the PYR cells, while VIP cells received a constant external input. In this case, sufficiently high PYR input and G value caused a jump back to higher SST activity in both layers (***Figure 7B***) with a sudden suppression of PV and VIP activity (***Figure 7—figure supplement 2A, B***). In contrast to the PYR cells, stimulation of VIP cells affected the oscillations and their power in a nonmonotonic fashion: the oscillation frequency increased initially, but dropped (superficial layer) or saturated (deep layer) at the transition to the SST state, while oscillatory power declined and suddenly increased with the switch to SST activity (***Figure 7—figure supplement 2C***). Finally, we also stimulated PYR cells while only PV cells received a constant external input. In this scenario, we obtained results similar to those obtained in the second scenario (***Figure 7C***, ***Figure 7—figure supplement 3A–C***). These findings are consistent with recent optogenetic experiments in which strong PYR drive was associated with high SST activity and comparatively low PV rates (***Hakim et al., 2018***).

Pyramidal cells also drive each other via recurrent excitatory connections, albeit such connectivity can be sparse (*Jiang et al., 2015*). To test the impact of locally recurrent excitation on the switching properties of the circuit, we systematically changed the connectivity between PYR cells in both layers (*Figure 7—figure supplement 4A*). With stronger weights, the sigmoidal SST response curve to VIP input was shifted to the right, while at sufficient strength network activity exploded (*Figure 7—figure supplement 4B*). Surprisingly, the switching dynamics was preserved even in the absence of recurrent excitation, highlighting the fact that the switch is implemented by interactions among the three inhibitory neurons types. The oscillation frequency in the high activation state of the VIP switch decreased with more recurrent excitation (*Figure 7—figure supplement 4C*).

## State changes propagate between superficial and deep layers

Next, we addressed the question whether a state transition triggered in only one layer propagates to the other layer across translaminar connectivity. To this end, we induced the same state changes as studied above (*Figures 3 and 4*) by applying current to specific inhibitory neurons in only the superficial or deep layer and measured the response of pyramidal cells in the opposite layer (*Figure 8*). In addition, we removed translaminar connections to test their role in the state propagation (only connections with an impact are shown). When the circuit displayed high SST activity, input to VIP cells in the superficial layer showed a disinhibitory PYR rate increase in the deep layer, which was mainly due to a translaminar reduction of SST activity rather than a direct drive from superficial to deep PYR cells (*Figure 8A*). We obtained a similar result in the superficial layer after driving the deep layer VIP cells (*Figure 8B*). Likewise, input to PV cells in the superficial layer caused disinhibition in the deep layer within a certain input range which was abolished by removing translaminar SST connections (*Figure 8C*). Notably, translaminar PV connections reduce the disinhibition effect as their removal strongly augmented PYR activity. The same effects were found in the superficial layer after PV input to the deep layer (*Figure 8D*). When we set the circuit to a VIP dominated state ($I_{ext}$ to VIP in both layers = 40Hz) and applied current to SST cells in the superficial layer, PYR cell activity was suppressed in the deep layer due to a direct translaminar SST connection (*Figure 8E*). The same results held true for SST input to the deep layer (*Figure 8F*). Finally, a similar suppressive effect on PYR cell rates that propagated to the other layer after input to SST cells was found in the presence of the PV dominated state ($I_{ext}$ to PV in both layers = 40Hz), again due to the translaminar SST connections (*Figure 8G, H*). In summary, these results demonstrate that transitions between disinhibited and inhibited states triggered in one layer can propagate to the opposite layer and this interlayer interaction is primarily governed by translaminar SST connections.

## Feedforward and feedback input

The above model results mimic circuit responses to optogenetic stimulation of specific cell types in living neuronal networks. However, visual stimulation of mouse V1 drives local circuits through feedforward and feedback drive. Superficial and deep layers receive feedback drive from different layers in higher areas, while feedforward input to superficial layers is provided by layer 4 of V1 circuits. The extent of this drive through PYR cells is cell specific, as for instance L4 preferentially drives PYR and PV cells in superficial layers (*Adesnik et al., 2012*), while feedback input from the anterior cingulate cortex mostly targets VIP cells (*Zhang et al., 2014*; *Figure 9A*).

To model the effects of feedforward and feedback inputs, we scaled the weight of external input to all cells according to the empirical data on feedforward and feedback inputs (*Adesnik et al., 2012*; *Zhang et al., 2014*; *Figure 9A*). For small $G$ values all cell types enhanced their activity for both types of drive, while we found a gradual suppression of SST cells for larger values (*Figure 9B, C*). This finding was more pronounced with feedforward input indicating stronger recruitment of inhibition. As the $G$ value increased, the rise of PYR cell activity was slowed or turned into suppression for the feedforward drive. In both input models we observed a suppression of the spontaneous low-frequency oscillation and a corresponding enhancement of higher frequency oscillations due to strong PV and VIP activation (*Figure 9D*). These results are consistent with the experimental literature, which reported alpha oscillation suppression and gamma oscillation enhancement upon both visual stimulation (*van Kerkoerle et al., 2014*) and feedback associated visual attention (*Klimesch, 2012*).

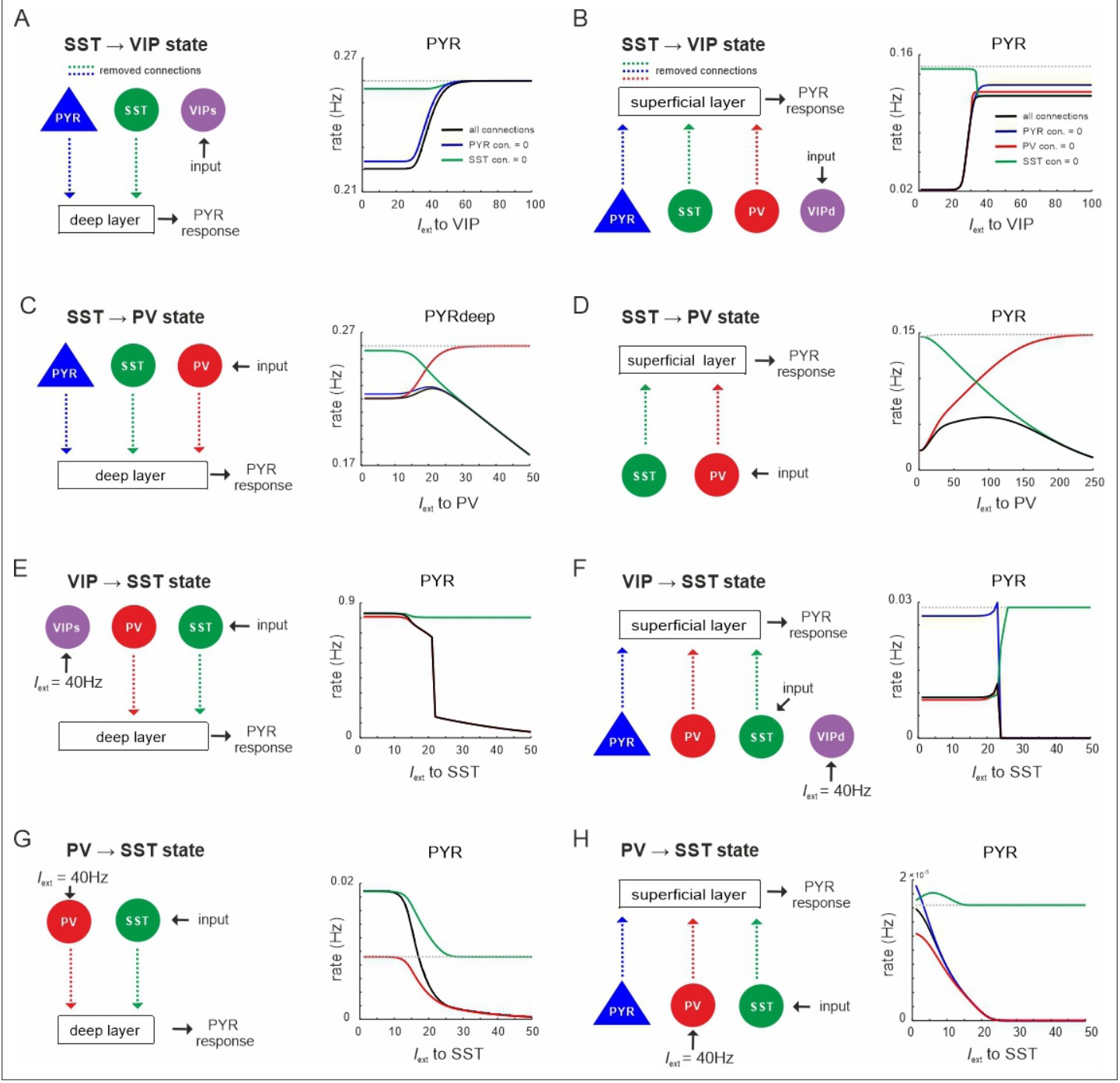

**Figure 8.** State changes propagate across layers. (**A**) Input was only given to vasoactive intestinal polypeptide (VIP) cells targeting the superficial layer and the PYR response was measured in the deep layer after all connection from superficial PYR or somatostatin (SST) cells to deep layer cells (dashed lines) were severed (left). The PYR response is shown for the case with intact connections (gray dashed line), all indicated connections cut (black solid line) or individual connections removed (colored lines). (**B–H**) Same as in (**A**) for input to different cells and different layers. Only connections were removed that had a visible influence on the PYR response as compared to the case where all connections were left intact.

## Recurrent inhibitory connectivity differentiates inhibited from disinhibited states

The connectivity between different interneuron types and pyramidal cells allows the microcircuit to switch between a disinhibited state with high PYR cell firing and an inhibited state with reduced PYR activity, as we show above. However, for a deeper understanding of this switch one important

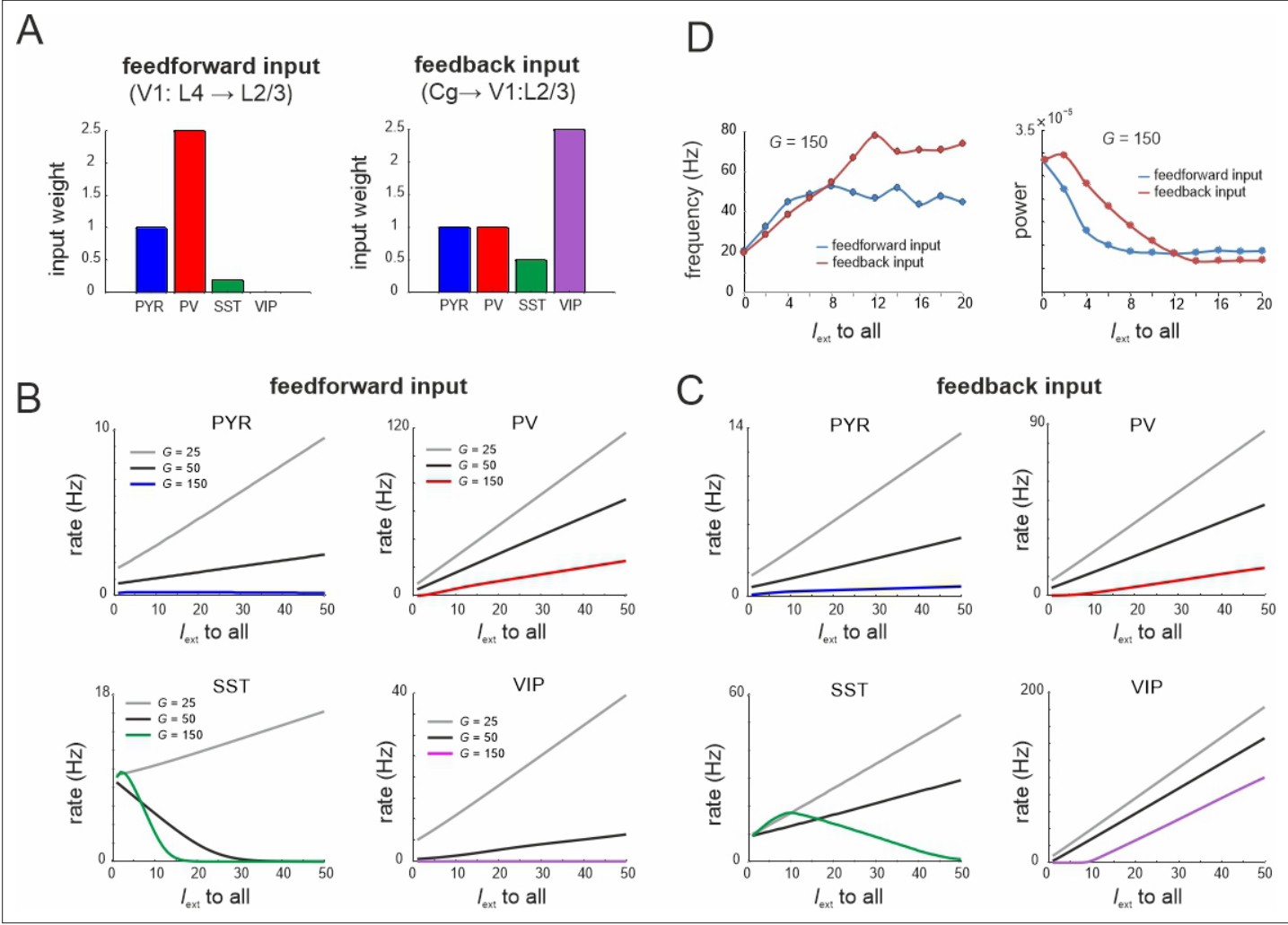

**Figure 9.** Microcircuit response to feedforward and feedback input. (**A**) Relative feedforward and feedback input weights of cell types normalized to PYR cells (weight = 1) with which external input $I_{ext}$ to all cells was scaled. (**A**) has been adapted from Figure 2C from **Adesnik et al., 2012** (left) and Figure 6D from **Zhang et al., 2014** (right). (**B**) Response of different cell types to input $I_{ext}$ scaled by feedforward weights and different values of G. (**C**) Same as in (B) for feedback input. (**D**) Peak oscillation frequency and power as a function of feedforward and feedback input strength.

question remains: Why are PYR cells more inhibited in the SST dominated regime as compared to the PV governed regime? To this end, we examined the mouse V1 connectivity matrix and found that there is a strong asymmetry between recurrent connections among PV and SST cells. While connections between PV cells in each layer are the strongest within the entire matrix, recurrent connections among SST cells are entirely absent (**Figure 10A**, **Supplementary file 1d**). Thus, we hypothesized that the strong self-inhibition of PV cells effectively reduces their inhibitory effect on PYR cells, while SST cells can elicit comparatively stronger inhibition on PYR cells due to the absence of self-inhibition. If this was true, exchanging the recurrent connections, that is removing PV self-connections and add them to SST cells may invert their role in microcircuit state switching (**Figure 10A**). In simulations with the inverted connectivity scheme we found that input to SST cells indeed created a disinhibition peak in the PYR cell response (**Figure 10B**, left), similar to VIP input in the original connectivity scheme (**Figure 3A**, **Figure 4—figure supplement 1B**), while PV and VIP cells showed a monotonic suppression (**Figure 10B**, right, **Figure 10—figure supplements 1 and 2A**). Likewise, driving VIP or PV neurons was followed by inhibition of all the other cell types, similar to the effect of SST input in the original case (**Figure 10C, D**, **Figure 10—figure supplement Figure 10—figure supplements 1 and 2B, C**). A notable exception is disinhibition in deep layer after PV input (**Figure 10—figure supplement 2C**). These results indeed suggest that inhibitory state-dependent alterations in excitation–inhibition

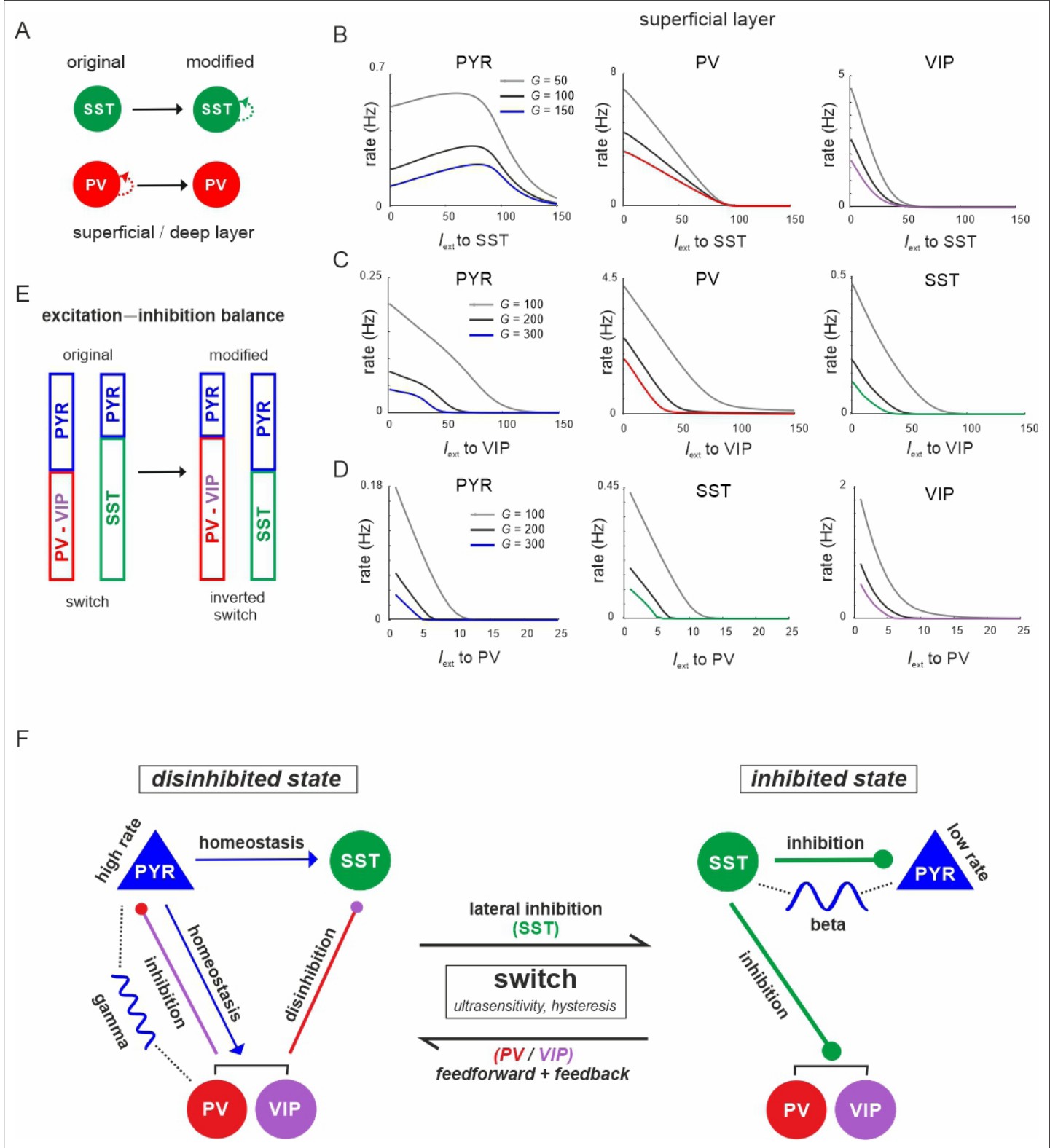

**Figure 10.** The microcircuit acts as a homeostatic switch between different levels of excitation and inhibition balance. (**A**) Schematic showing the typical pattern of strong recurrent connections between parvalbumin (PV) cells and their absence in somatostatin (SST) cells in superficial and deep layers (left). The pattern was inverted in the modified connectivity matrix (right). (**B**) Response of PYR, PV, and vasoactive intestinal polypeptide (VIP) cells to SST input using the modified connectivity matrix and three different values of G. Note the increase in PYR activity with moderate SST input. (**C**) Same as in (**B**) for input to VIP cells and response of PYR, PV, and SST cells. (**D**) Same as in (**A**) for response of PYR, SST, and VIP cells to PV input. (**E**) Schematic showing

*Figure 10 continued*

how the excitation–inhibition balance changes between PV/VIP and SST dominated states using the original (left) or modified connectivity matrix, as shown in (**A**). Excitation is mediated by PYR cells and inhibition by VIP/PV or SST cells. (**F**) Summary diagram displaying the principles of the homeostatic switch implemented in the connectivity matrix of the studied microcircuit.

The online version of this article includes the following figure supplement(s) for figure 10:

**Figure supplement 1.** Response of somatostatin (SST), parvalbumin (PV), and vasoactive intestinal polypeptide (VIP) cells to input to the same cell type for three different values of *G* in the superficial layer after exchange between PV and SST recurrent connectivity.

**Figure supplement 2.** Responses to input to different cells after exchange between parvalbumin (PV) and somatostatin (SST) recurrent connectivity.

balance, which are at the core of the switching properties of the microcircuit, are due to large differences in PV and SST self-connectivity (*Figure 10E*).

## Discussion

In this study, we showed that superficial and deep layers of the mouse V1 microcircuit are endowed with two switch mechanisms that can toggle the pyramidal cells (output of the microcircuit) between high (disinhibited) and low activity states (inhibited) across superficial and deep layers (*Figure 10F*). The underlying switching mechanics are realized by the interactions among the three interneuron types (PV, SST, and VIP), which compete for inhibitory influence on pyramidal cells. In the inhibited state, SST cells dominate inhibition which serve as 'master regulators' by strongly connecting and inhibiting activity of pyramidal cells and other interneurons in the circuit (*Jiang et al., 2015*; *Chen, 2017*; *Fino and Yuste, 2011*; *Urban-Ciecko and Barth, 2016*; *Naka et al., 2019*). In the disinhibited state, excitation is mainly balanced by PV cells (*Ferguson and Gao, 2018*) and to a lesser extent by VIP neurons (*Garcia-Junco-Clemente et al., 2017*), whereas SST activity is reduced.

### Difference between inhibition exerted by PV and SST neurons

While disinhibition through SST suppression was previously shown experimentally and theoretically (*Pi et al., 2013*; *Walker et al., 2016*; *Hertäg and Sprekeler, 2019*; *Wang and Yang, 2018*), the question remains why SST cells provide more inhibition than PV or VIP cells, even though the weights of the PV to PYR connections by far outweigh SST to PYR connections (*Supplementary file 1d*). Similar to previous simulations of anatomically simplified neuronal networks (*Hertäg and Sprekeler, 2019*), we found that a key to this inhibitory asymmetry lies in the degree to which PV and SST cells are connected among themselves. While SST cells lack mutual connectivity, PV cells have strong mutual connectivity (that may also be reinforced by gap junctions *Pfeffer et al., 2013*; *Jiang et al., 2015*; *Bartos et al., 2007*). In our rate model, self-inhibition of PV cells reduced their impact on PYR cells and enhanced PYR rates. We note that a similar effect was found in spiking network models, where PV interactions enhanced PYR rates and synchrony (*Hahn et al., 2014*). Accordingly, we found that exchanging self-connections of PV and SST neurons inverted their role in mediating the inhibited or disinhibited state.

### Origin of switching dynamics

The switching mechanism emerges as a consequence of two mutual inhibitory connection motifs that is from SST to PV and VIP, and PV/VIP to SST neurons, in both layers. Thus, driving SST cells switches the microcircuit state to the 'inhibited state' by suppressing PV and VIP neurons as well as PYR neurons. Conversely, input to VIP or PV cells disinhibits the circuit by decreasing SST cells activity. The nature of this switch depended on the mutual inhibitory connectivity strength, rendering the state transition purely ultrasensitive with sigmoidal input–output curves for weaker weights or bistable with hysteresis and memory for stronger weights, consistent with recent observations made in simplified inhibitory neuronal networks (*Hertäg and Sprekeler, 2019*). Switching based on double negative feedback has been conserved during evolution in many biological systems (*Ferrell and Ha, 2014*; *Ferrell, 2002*; *Tyson et al., 2003*) and we found that it may also be a hallmark of cortical microcircuits. Notably, the two ways to control the switch differ in their sensitivity, with VIP cells requiring considerably less input to control SST activity and flip the circuit to the disinhibited state in both layers than PV cells. Thus, PV cells seem to be more specialized in keeping the excitation–inhibition balance at

sufficiently high levels, whereas VIP cells play more a switching role, even though both cell types can assume both functions.

The switching circuitry of the microcircuit can not only be activated by simultaneous input to superficial and deep layers, but each layer can transmit its state change to the other through translaminar connectivity, and effectively synchronize inhibited or disinhibited states across the whole microcircuit. These results suggest that the two-layer microcircuit as a whole can act as a switch, whose state may be used to guide activity flow of pyramidal cells in the feedforward (via L2/3) and feedback direction (L5) (*Markov et al., 2014b*; *Markov et al., 2014a*). However, it is conceivable that differential input to superficial and deep layers could place both in different switch states.

## Bistable dynamics

The presence of bistability based on mutual inhibitory connection motifs also provides an alternative to the dominant view that persistent transitions between low and high firing rate states across the brain (*Christophel et al., 2017*), which underlie working memory required for attention (*Ardid et al., 2014*), consciousness (*Dehaene et al., 2014*; *Dehaene and Changeux, 2011*; *van Vugt et al., 2018*), and language processing (*Kunze et al., 2017*; *Nelson et al., 2017*; *Pallier et al., 2011*), are primarily based on local (*Compte et al., 2000*) or interareal (*Dehaene et al., 2014*; *van Vugt et al., 2018*) recurrent connectivity between excitatory neurons. In our study, bistability is controlled by properties of local inhibitory connectivity, opening up the possibility that anatomical heterogeneity and gradients across the brain may reflect the presence of bistable switches in some brain areas and ultrasensitive switches without memory in other regions (*Kim et al., 2017*; *Fulcher et al., 2019*; *Demirtaş et al., 2019*).

## Oscillations in the microcircuit

Two negative feedback loops between pyramidal cells and interneurons endow both deep and superficial layers with an intrinsic oscillation. Consistent with experiments and models, PV cells drive high frequencies ('gamma range'), whereas SST cells prefer lower frequencies ('beta range') (*Cardin et al., 2009*; *Chen, 2017*; *Lee, 2018*). Accordingly, during the inhibited switch state the dominating SST cells impose a slower frequency on PYR cells as compared to the PV dominated disinhibition state with high frequencies. VIP cells had little influence on oscillation frequency. An interesting finding of our study is that this switch in frequency is already shaped by the connectivity pattern of the circuit and is only partly modified by intrinsic neuronal properties such as differing time constants. Note also that in the absence of noise and oscillations, the ability of the network to change its frequency is not dependent on the exact time constant of neuron populations suggesting that the switch is already engrained in the connectivity. Neuron population time constants, however, do determine the amount of change in the oscillation frequency as the network goes from SST to PV mode. This result thus provides a potential explanation for a longstanding question of what determines the oscillation frequency in neuronal circuits (*Brunel and Wang, 2003*; *Jia et al., 2013*). This finding is in contrast to previous modeling efforts which introduced slow and high frequencies by endowing interneurons with different time constants (*Hyafil et al., 2015*; *Mejias et al., 2016*). Animal experiments have revealed that the power of high-frequency oscillations was stronger in the superficial layer, while slower oscillation had more power in the deep layers (*van Kerkoerle et al., 2014*; *Haegens, 2015*; *Buffalo et al., 2011*; *Bonaiuto et al., 2018*). Based on our results we argue that the emergence of such differences in oscillations in deep and superficial layers crucially depends on specific anatomical connections (see *Figure 2A*). Moreover, our results also replicate experimental findings which have shown that visual stimulation mediated by feedforward connections and visual attention mediated by feedback connections suppress slow oscillations (alpha band) and enhance fast oscillations in the gamma band (*van Kerkoerle et al., 2014*; *Klimesch, 2012*).

Thus, our results also suggest an alternative, anatomically based taxonomy for oscillations, dividing them into competing SST- (low frequency) and PV-driven rhythms (high frequency). Notably, during the transition between (PV or SST dominated) switch states the oscillation frequency and power behave asymmetrically, that is an increase in frequency is accompanied by a decrease in power. However, when PV or SST cells are strongly suppressed or silenced, frequency and power change symmetrically. A decrease in frequency and increase in power is also found in recent experiments in monkeys and mice, in which small stimuli trigger a high-frequency oscillation (~60 Hz) that is replaced by a lower

frequency (~30 Hz) in the presence of larger stimuli (*Chen, 2017*; *Veit et al., 2017*; *Gieselmann and Thiele, 2008*).

Anatomical studies show that this surround suppression effect is presumably mediated by lateral excitatory input from the surround to SST neurons in the center (*Adesnik et al., 2012*). Our model suggests that the surround switches the microcircuit from a PV/VIP dominated state with a high frequency, putatively mediated by stimulus triggered feedforward and feedback drive (*Zhang et al., 2014*; *Gonchar and Burkhalter, 2003*), to a more inhibited state caused by enhanced lateral drive to SST cells. We note that experiments have also reported the inverse case with an increase in oscillation frequency and decrease in power after enhancing visual stimulus contrast (*Ray and Maunsell, 2010*).

The switching properties in the model depend on the coupling parameter $G$, which scales the overall connectivity between neurons of the circuit. This parameter has been previously used to study whole-brain models of cortical dynamics (*Deco et al., 2013*) and can be related to the overall cell count in the studied circuits. The experimentally given connectivity matrix used here (*Jiang et al., 2015*) only describes the connection strength between pairs of neurons and therefore it becomes necessary to scale the weights while investigating the dynamics of the firing rate-based model. As the coupling parameter critically determines the presence of ultrasensitivity, bistability, and oscillation properties, it is crucial for further studies to determine its value, which may differ between different circuits across the brain. We found that feedforward and feedback input lead to suppression of SST and PYR activity for sufficiently high $G$ values. This is inconsistent with recent experimental results in mouse V1, where the rate of all cell types increased after visual stimulation (*Dipoppa et al., 2018*; *Kirchberger et al., 2021*). This suggests that smaller values of $G$ may be biologically more realistic for mouse V1.

## Anatomical model limitations

The connectivity matrix used in the model is based on the detailed anatomical study of *Jiang et al., 2015*, which did not capture layers 4 and 6 of mouse V1. There exist a number of studies which have charted the different morphological and biochemical cell types in these two layers, which can also be divided into PV, SST, and VIP classes similar to layers 2/3 and 5 in the present study (*Scala et al., 2019*; *Frandolig et al., 2019*; *Yavorska and Wehr, 2021*; *Ding et al., 2021*). However, there is limited anatomical knowledge about the cell-specific connectivity within each layer and their connections to other layers in the microcircuit. We therefore excluded these layers from our study and restricted our model to the more comprehensively mapped connectivity of layers 2/3 and 5. It is possible that cell type-specific connectivity within the layers 4 and 6 and feedback from layer 6 to the thalamus can change the network dynamics, but that requires more detailed investigation.

The study of *Jiang et al., 2015* also included layer 1, which contained only two inhibitory cell classes. One of them, neurogliaform cells, was also found in the other layers and provides inhibition to all other cells presumably through volume transmission, which is not suitable for our modeling approach. The second type, single-bouquet cell-like neurons, targets VIP cells in the superficial layer and were not considered in the model.

## Stability of the cortical microcircuit

The disinhibited state of the switch naturally entails the risk of runaway excitation. Our results provide evidence that the microcircuit contains supplementary homeostatic mechanisms that keep disinhibition within healthy boundaries. When the circuit is disinhibited and SST activity suppressed, strong drive to pyramidal cell can cause a sudden reversal of SST neurons to a high firing rate state and thereby restore a more inhibited state in the microcircuit. This phenomenon was reported experimentally (*Pakan et al., 2016*) and in a recent theoretical study (*Garcia Del Molino et al., 2017*) showing elevated visually evoked SST rates, after prior suppression through VIP cells during darkness.

## Related experimental data from nonhuman primates

While our results were obtained using anatomical data from mouse V1, the model also replicates some of the experimental findings obtained from monkey V1 recordings. Notably, the prevalence of gamma oscillations in the superficial layers and stronger power of slow frequency oscillations in deep layers has been reported by several studies (*van Kerkoerle et al., 2014*; *Haegens, 2015*; *Buffalo et al., 2011*; *Bonaiuto et al., 2018*). Moreover, the abrupt transition from high frequency to slow frequency

oscillations due to increase in visual stimulus size and associated surround suppression size is a classic finding in monkey V1 (*Gieselmann and Thiele, 2008*). Despite the lack of cell type-specific anatomical knowledge in the monkey primary visual cortex (*Vanni et al., 2021*), these results suggest that the canonical microcircuit of both species may share common principles.

## Predictions

We found that asymmetric changes in oscillation frequency and power are abolished with silencing of PV or SST cells, a result that can be experimentally tested. Moreover, our model predicts that driving PYR cells is followed by linear or nonlinear responses in inhibitory cells depending on the dominance of SST or PV/VIP cells, respectively. Finally, experiments could test whether inhibition or disinhibition mediated by driving specific inhibitory cells in one layer propagate to the other layer, as our simulations showed.

In conclusion, the microcircuit function that emerges from our computational study is a homeostatic switch that toggles pyramidal cells, the principal output neurons of the circuit, between an inhibited and disinhibited state. The switching dynamics is orchestrated by an array of inhibitory neurons, each performing a specific task in the switch mechanics. Feedforward, feedback, and lateral input may change the position of the switch and regulate the flow of excitation to downstream microcircuits (*Cardin, 2018*; *Fishell and Kepecs, 2020*; *Hahn et al., 2021*). Our results also map different types of oscillations onto different interneuron types and link them with distinct switch states, which in the future may help to bring together rate- and oscillation-based experimental paradigms. They also provide mechanistic insight into the long held notion that slow oscillations assume an inhibitory function, while fast oscillations serve information processing (*Hahn et al., 2021*; *Klimesch et al., 2007*; *Jensen and Mazaheri, 2010*; *Singer, 1999*; *Fries, 2015*).

## Methods
### Microcircuit architecture

In this study, we developed a firing rate model of the visual cortical microcircuit. This model is based on pairwise connectivity between major neuron types in superficial and deep layers of the neocortex (*Jiang et al., 2015*). Using octuple whole-cell recordings, *Jiang et al., 2015* exhaustively mapped out the connectivity (excitatory postsynaptic potential (EPSP) and inhibitory postsynaptic potential (IPSP) strength and connection probability) between a large number of morphologically defined neurons (interneurons and pyramidal cells) within and between layers 1, 2/3 (superficial), and 5 (deep) of the primary visual cortex of the mouse, while layers 4 and 6 were not included. In addition, *Jiang et al., 2015* also quantified the relative prevalence of each cell type and reported synaptic plasticity properties of specific connections. Finally, based on their genetic makeup, different interneurons were labeled as PV cells (layer 2/3: basket cells and chandelier cells; layer 5: shrub cells and horizontally elongated cells), SST cells (layers 2/3 and 5: Martinotti cells), and VIP cells (only layer 2/3: bitufted cells and bipolar cells).

To convert the original connectivity data (see *Supplementary file 1a* for the connection probability matrix and *Supplementary file 1b* for the weight [EPSP/IPSP] matrix) into a format suited for a computational model, we performed several manipulations. First, restricting our analysis to pyramidal cells and three major interneurons types (PV, SST, and VIP), we created a single connectivity matrix ($C$) by element-wise multiplication of EPSP/IPSP strength (i.e., mean amplitude) and connection probability matrices. To generate a single class of PV cells in each layer, we added the weights of basket cells and chandelier cells in the superficial layer and the weights of basket cells, shrub cells, and horizontally elongated cells in the deep layer. After inspection of the connectivity matrix, we noticed that out of the two existing VIP cell types, one VIP cell type preferentially targeted other neurons in L2/3 (bitufted cells), whereas the other only innervated L5 (bipolar cells). Based on this observation, we divided VIP neurons into $VIP_{sup}$ and $VIP_{deep}$ cell types, even though both types were anatomically located in the superficial layer.

Thus, we had four neuron types in both deep and superficial layers. Instead of modeling populations of neurons with $N$ cells for each class, we modeled each cell population using a single firing rate-based neuron. However, we adapted the connectivity weights according to their relative prevalence, which is computationally less expensive. To this end, we first followed the general rule that there are

roughly five times more pyramidal neurons in a microcircuit than interneurons. Therefore, all inhibitory weights in the matrix were scaled down by a factor of 0.2. Next, we multiplied the weights of each interneuron type with its relative prevalence (*Supplementary file 1c*). This resulted in the corrected 8 × 8 connectivity matrix (*Supplementary file 1d*). The resulting microcircuit is schematically shown in *Figure 1A*.

In addition, for the analysis in *Figure 4—figure supplement 2B, C*, we jittered the connectivity matrix (50 trials) by adding values that were drawn from a standard Gaussian distribution and scaled by a factor of 0.2, in accordance with the experimental literature (*Jiang et al., 2015*).

## Population activity model

The dynamics of each neuron type, that is pyramidal cells and interneurons, was modeled using a coarse-grained firing rate-based model (Wilson–Cowan model [WC], *Wilson and Cowan, 1972*). The dynamics of the full microcircuit can be written in vector form as:

$$\tau \frac{dr}{dt} = -r + \varphi \left( I_{\text{circuit+I}_{\text{ext}}} \right) + \eta \tag{1}$$

where $r$ is the vector of rates of all eight cell types, $\tau$ is the vector of neuron-specific time constants (*Supplementary file 1e*), and $\eta$ reflects Gaussian noise with standard deviation $\sigma = 0.01$. In the Wilson–Cowan type firing rate model $\tau$ refers to the time constant of the neuron population and it cannot be directly compared with the synaptic or membrane time constants. $\tau$ can be experimentally inferred by measuring the time constant of neuronal spiking activity in ongoing or evoked states. Often such measurements are restricted to the time constant of pyramidal neurons as their activity is more likely to be picked up by extracellular electrodes.

The input–output firing rate transfer function ($\varphi$) of each neuron type was modeled as $\varphi(x)\frac{ax}{1-e^{-x/b}}$ and is identical for all neuron types. The parameter $a$ controls the slope of the transfer function and the parameter $b$ controls the threshold. The parameters $a$ and $b$ were set to 1 for most analysis and varied to generate *Figure 4—figure supplement 2D, E*. We followed the approach of *Mejias et al., 2016* and modeled each neuronal population with a single WC equation. Note however that despite this simplification our results on the switching dynamics and transition between different frequency bands remain qualitatively similar to studies that have used either multiple WC neurons per cell type (*Hertäg and Sprekeler, 2019*) or implemented networks of multiple interneuron types with spiking neuron models (*Lee, 2018*; *Domhof and Tiesinga, 2021*). The term $I_{\text{circuit}}$ denotes the input from other neuron populations across the entire microcircuit and is given by:

$$I_{\text{circuit=Cr}} \tag{2}$$

where $C$ is the corrected connectivity matrix. $I_{\text{ext}}$ reflects external input to different neuron populations from bottom-up, top-down, and lateral connections. Due to strong inhibition, PYR activity can be reduced to zero and even though the network is oscillating we may not observe those, as our readout of network oscillations is through PYR neurons. Therefore, it was necessary to inject some external input to PYR neurons $I_{\text{ext}}$ to PYR = 5 Hz (unless stated otherwise) in order to observe oscillations. The time constant $\tau$ was initially chosen for each cell type in accordance with the experimental and model literature, which attributed a fast decay to PYR and PV activation and a longer decay to SST cells (*Chen, 2017*; *Hyafil et al., 2015*; *Mejias et al., 2016*; *Silberberg and Markram, 2007*; *Vierling-Claassen et al., 2010*). However, we also varied the time constants for each cell type to test their influence on oscillatory switch dynamics and create *Figure 5E–H*. Because distances within the microcircuit are very short, conduction delays were not modeled explicitly. For the analysis of the rate response of a given cell class to input, we performed a noise free ($\sigma = 0$) simulation of 2 s after which a steady-state response was reached. The rate was used for further analysis (*Figures 3 and 5–7*). For *Figure 1C*, 20 s were simulated, and the rate was averaged across the entire duration. To create and study oscillations, we added noise to the neurons ($\sigma = 0.01$), simulated 50 trials of 20 s each and averaged oscillation metrics across all trials. All simulations were performed with a time step of 0.1 ms.

## Control variables

To study the behavior of the microcircuit, we varied several parameters. Foremost, we studied the dynamics of the microcircuit by systematically changing the overall connectivity by the scaling factor $G$. The scaling of the connectivity matrix can be loosely related to altering the absolute number of

neurons in our circuit, similar to other studies (*Deco et al., 2013*; *Jobst et al., 2017*). For the analysis, we used the entire connectivity matrix without masking weak connections. The behavior of the circuit was studied by manipulating individual connections, removing or swapping them. Optogenetic silencing of individual cell types was simulated by setting specific columns of the matrix to zero, that is we effectively removed all output of specific neuron types to the entire microcircuit (see *Figure 2*). To mimic stimulation of specific neuron types, we directly injected current with varying amplitudes to the selected neuron type (e.g., see *Figures 3–6*).

## Data analysis
### Analysis of the LFP
We used the rate of pyramidal cells in each layer as a proxy for superficial and deep LFPs and its oscillatory behavior was investigated as a function of external drive. To analyze oscillation in the LFP, we computed the power spectrum within a range of 1–250 Hz with the multitaper method implemented in the Chronux toolbox of Matlab (http://chronux.org/). The power spectrum was smoothed and normalized by the summed power of the computed frequency range. We then quantified visible oscillation peaks, excluding frequencies <10 Hz, in terms of peak frequency and power.

### Measure of switching dynamics and hysteresis
Ultrasensitivity reflects the behavior of a system where small changes in input cause large changes in output. Such behavior is observed in many natural systems such as biochemical reactions (*Ferrell and Ha, 2014*; *Ferrell and Machleder, 1998*). Ultrasensitivity can be quantified by fitting a sigmoidal curve to the input–output transfer function. Here, we used the Hill equation (*Ferrell and Ha, 2014*) to estimate the sensitivity of the output ($y$) to the input ($x$):

$$y\left(x\right) = c + \frac{dx^n}{k^n + x^n} \tag{3}$$

where $c$ is the intercept, $d$ is the maximum, $k$ is the half-maximum of the output $y$ and $n$ is the Hill exponent (also denoted as $n_H$ in the results section), which we used to quantify the response curves of SST cells to VIP and PV input (*Figure 4A–C*). If $n = 1$, the Hill curve is hyperbolic, whereas $n > 1$ indicates a sigmoidal shape with growing slope (i.e., sensitivity) as $n$ increases.

Hysteresis in general describes the dependence of a system's behavior on the past and implicates the presence of memory. As a consequence, system responses observed when input is steadily increased differ from responses to decreasing input. To test for hysteresis, we computed response curves of all cells (*Figure 3C, D*, *Figure 3—figure supplement 4*) for ascending and descending VIP and PV input separately. For each input value ($i$), the rate response of all cells of the previous input value ($i − 1$) was used to initialize the cell rates for the new input value. In the presence of hysteresis, the response curves for increasing and decreasing input do not collapse. Hysteresis ($h$) was then quantified as the summed difference of the ascending and descending rate curves of the SST cells in superficial and deep layers.

## Acknowledgements
G.H., H.S., T.R.K., and G.D. were funded by the German Research Council (DFG, No. KN 588/7-1) within the priority program Computational Connectomics (SPP 2041). G.D. was supported by the Spanish Research Project AWAKENING: using whole-brain models perturbational approaches for predicting external stimulation to force transitions between different brain states, ref: PID2019-105772GB-I00 / AEI/10.13039/501100011033, financed by the Spanish Ministry of Science, Innovation and Universities (MCIU), the Spanish Research Project COBRAS PSI2016-75688-P (AEI/FEDER, EU), and by the Catalan AGAUR program 2017 SGR 1545. G.D. and G.H. received support from the European Union's Horizon 2020 research and innovation program under Grant Agreement No. 720,270 (Human Brain Project SGA1) and No. 785,907 (Human Brain Project SGA2). G.H. was funded by the grant CONSCBRAIN (n. 661583) of the European Union's Horizon 2020 research and innovation program under the Marie Skłodowska-Curie action. A.K. received support from the Swedish Research Council.

## Additional information

### Funding

| Funder | Grant reference number | Author |
|---|---|---|
| German Research Council | SPP 2041 | Gerald Hahn<br>Helmut Schmidt<br>Thomas R Knösche<br>Gustavo Deco |
| Spanish Research Project COBRAS | PSI2016-75688-P | Gustavo Deco |
| Catalan AGAUR program | 2017 SGR 1545 | Gustavo Deco |
| Horizon 2020 - Research and Innovation Framework Programme | 720270 | Gerald Hahn<br>Gustavo Deco |
| Horizon 2020 - Research and Innovation Framework Programme | 785907 | Gerald Hahn<br>Gustavo Deco |
| Horizon 2020 - Research and Innovation Framework Programme | 661583 | Gerald Hahn<br>Gustavo Deco |
| Swedish Research Council | | Arvind Kumar |
| Max Planck Institute for Human Cognitive and Brain Sciences | open access funding | Helmut Schmidt<br>Thomas R Knösche |
| Spanish Ministry of Science, Innovation and Universities | PID2019-105772GB-I00 / AEI/10.13039/501100011033 | Gustavo Deco |

The funders had no role in study design, data collection, and interpretation, or the decision to submit the work for publication.

### Author contributions

Gerald Hahn, Conceptualization, Software, Formal analysis, Investigation, Visualization, Methodology, Writing - original draft, Writing – review and editing; Arvind Kumar, Conceptualization, Investigation, Methodology, Writing – review and editing; Helmut Schmidt, Conceptualization, Methodology, Writing – review and editing; Thomas R Knösche, Conceptualization, Supervision, Funding acquisition, Methodology, Writing – review and editing; Gustavo Deco, Conceptualization, Supervision, Funding acquisition, Writing – review and editing

### Author ORCIDs

Gerald Hahn http://orcid.org/0000-0002-7069-0639
Arvind Kumar http://orcid.org/0000-0002-8044-9195
Helmut Schmidt http://orcid.org/0000-0002-2264-0821
Thomas R Knösche http://orcid.org/0000-0001-9668-3261

### Decision letter and Author response

Decision letter https://doi.org/10.7554/eLife.77594.sa1
Author response https://doi.org/10.7554/eLife.77594.sa2

## Additional files

### Supplementary files

• Supplementary file 1. Anatomical connectivity parameters. (a) Connection probabilities for different morphologically defined cell types, as described in *Jiang et al., 2015*. BC: basket cells; ChC: chandelier cells; MC: Martinotti cells; BTC: bitufted cells; SC: shrub cells; HEC: horizontally elongated cells; BPC: bipolar cells. (b) EPSP/IPSP strength for different morphologically defined cell types, as described in *Jiang et al., 2015*. BC: basket cells; ChC: chandelier cells; MC: Martinotti

cells; BTC: bitufted cells; SC: shrub cells; HEC: horizontally elongated cells; BPC: bipolar cells. (c) Morphological interneuron types, their genetic marker and proportion. (d) Connectivity matrix corrected for cell proportions and scaled up by $G$ = 100. (e) Time constants for different neuron types.

• Transparent reporting form

• Source code 1. The code provides ascending and descending input to vasoactive intestinal polypeptide VIP cells and plots the response of all cell types in superficial and deep layers, as shown in *Figure 3*.

• Source code 2. This code provides a function that implements the Wilson–Cowan model used to simulate the dynamics of each neuron population.

## Data availability

The current manuscript is a computational study and thus no data have been generated for this manuscript. The empirical connectivity matrix of the microcircuit is provided in Supplementary File 1d. Modeling code is uploaded as Source Code Files 1 and 2.

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
