## [Editor Report]

The microcircuit has a canonical composition and the interactions among distinct classes of excitatory and GABAergic neurons are fundamental to our understanding of sensory processing and neuronal synchronization. The authors investigate emerging dynamics in laminar models of the visual cortex, consisting of distinct GABAergic cell types, with a connectivity model based on the latest anatomical findings. The authors identify bistable circuit switches emerging from the interactions between different cell types and these are characterized by inhibited and disinhibited states accompanied by low- and high-frequency oscillations, respectively. These findings suggest a canonical, non-linear circuit motif that can explain multiple experimental observations and adds significantly to our understanding of microcircuit dynamics.

---

## [Decision Letter]

**Decision letter after peer review:**

[Editors’ note: the authors submitted for reconsideration following the decision after peer review. What follows is the decision letter after the first round of review. Please note the reviewers have opted to remain anonymous. ]

Thank you for submitting your work entitled "Computational Properties of the Visual Microcircuit" for consideration by *eLife*. Your article has been reviewed by 3 peer reviewers, one of whom is a member of our Board of Reviewing Editors, and the evaluation has been overseen by a Reviewing Editor and a Senior Editor. The following individual involved in review of your submission has agreed to reveal their identity: Thilo Womelsdorf (Reviewer #2).

Our decision has been reached after consultation between the reviewers. Based on these discussions and the individual reviews below, we regret to inform you that your work will not be considered further for publication in *eLife*. Although the reviewers expressed interest in the study, with some mixed enthusiasm, major issues were identified. Reviewers agreed that these preclude publication in *eLife* at present and would require a substantial amount of additional work to improve the manuscript. The main issues identified were the lack of input layers in the model, the dynamics of different neuron types and their impact on the system, and the idealization of the circuit as a single column. If you are able to address these points and the other points raised by the reviewers, *eLife* would welcome re-submission of this paper: This would be treated as a new submission, but would likely go to the same reviewers if it were to be sent out for review.

*Reviewer #1:*

In this study, the authors build a microcircuit of mouse primary visual cortex using connectivity data between four cell types (PV, SSt, Pyr, VIP) in two laminar compartments (superficial and deep). The authors then examine the influence of (1) Global increase in coupling (G), and (2) Specific drive to particular neurons. This basic microcircuit appears to capture some basic features of physiology, namely: (1) Increased high-frequency power in superficial layers and increased low-frequency power in deep layers (Figure 1), (2) Enhanced firing rates in deep layers. The main finding of the authors is that when they enhance drive to the VIP cells, there is a steep transition in the firing of SST cells, which they characterize as a bi-stable system. Overall, this study has merit in providing a quite extensive characterization of a canonical microcircuit, and the authors do a great job in linking their manipulations to experimental findings.

Concerns:

1) The authors describe the effect of driving VIP interneurons as disinhibitory. I'm confused. Figure 3 shows that the effect of driving VIP interneurons in pyramidal firing rates in superficial layers is largely suppressive, which seems to go against their main conclusion and experimental findings.

2) Noisy oscillations should be better defined. Are these noisy limit cycles or quasi oscillations?

3) The model provided by the authors has a stochastic drive and therefore exhibits noisy oscillations. It should be discussed whether similar results would have been obtained when using stochastic Wilson-Cowan models where each population consists of multiple neurons and the neurons have spiking output.

4) A limitation of the study is that Layer 4 is not modelled, and that the application of external drive appears somewhat arbitrary. In reality the external drive on PV and Pyramidal cells will likely be matched/balanced to some extent. However, Figure 5 applies a constant input to either PV or VIP cells which does not seem a realistic assumption. It would be hard to make reasonable assumptions on how the input drive to VIP cells looks like.

5) Related to this, the authors identify that the bi-stable switch appears to emerge in a particular regime of G. However, do we have any idea what the value of "G" in an actual circuit would be? Furthermore, should one not examine how the strength of external drive modulates the ability of the circuit to switch?

6) A very relevant study, Di Poppa et al. by Ken Harris lab, is only marginally discussed in this paper. However, that paper also provided a model of interactions between these cell classes and reports many findings on disinhibition and covariation between the cell types.

7) The authors should stress that this is a model of the MOUSE primary visual cortex. We know that the monkey visual cortex has quite different properties in terms of cell types, connectivity, lamination etc. It's unclear whether the mouse work bears any relevance for the monkey. The authors mix references to mice and monkey studies, but this should be made clear at all point. For instance, γ oscillations in monkeys have quite different signatures from the mouse in terms of frequencies. We can't really compare those in a straightforward way.

8) The study relies on the octo-patch data. However, that dataset was criticized quite severely in a commentary. Can references be made to other studies e.g. from Scanziani confirming some of these connectivity patterns?

*Reviewer #2:*

This is an excellent, innovative, well written and comprehensive modeling study. The paper is made possible with detailed anatomical data that is only available recently about connectivity matrices and cell type distributions across layers. The study systematically shows how drive to VIP (feedback type) and PV (feedforward type) cell modulate a SST cell dependent inhibited and a PV dependent disinhibited state. The study mechanistically tracks down the source for the two separable states asymmetric self-inhibition and show how state transitions can be induced ('toggled'). The paper provides strong quantifiable predictions which is attractive for empirical people.

A weakness is that many assumptions are implicit and key findings are difficult to find in the large amount of data.

1. It is difficult to get a good understanding which G values are realistic for invite states and what they mean. There is a rate and power change for G of < 100 (Figure 1), but then various interesting effects are occurring at G >200 (effects of cell silencing).

What are the underling characteristics / mechanisms that determine g changes, i.e. changes in effective coupling strength? describing this more explicitly early in the results would enhance comprehensibility

2. The presentation of the key contributions of SOM and PV is ideally improved. Going through Figure 2 is highly interesting but time consuming because the many panels make this a dense figure. Can the two key results of the effects of PV silencing and SOM silencing be shown as a power vs frequency plot – identical to Figure 2C and D. So far, we only see the peak frequencies across G but this does not easily allow to appreciate the power spectral shape at a low (e.g. 50) or high (e.g. 450) G. In its current form it is not possible to confirm how wide the power spectral density peaks are.

3. The results are describing average effects and are surprisingly non-quantitative. There is not a single statistic used and error bars are lacking. While this seems justified in many cases it leaves the impression that there is no noise in the results and that they would be perfectly reproducible and robust across may conditions. How realistic is this and what would change if realistic noise is introduced?

When the authors address this question, it would be important to see an explicit definition of what a 'state' refers to. The paper describes many simulations where inputs are switched on constantly and with no apparent fluctuations. Is that realistically happening in real feedforward /feedbacK signal dynamics?

*Reviewer #3:*

Using a Wilson-Cowen model, the authors look at the coupling between 8 neuronal populations (excitatory + 3 inhibitory) based on experimental connectivity of the L2/3 and L5 populations. They do a very thorough analysis of the oscillatory properties of this system under a set of assumptions. Even though they do not address computations directly, the results are interesting, and link to several hypotheses in the field on the generation of oscillatory behavior. However, I am not convinced that following the choices of parameters and approximations the model is a useful approximation of the biological circuit. I believe the authors need to do considerably more work to justify the choices of parameters and the implicit approximations used (or that they are irrelevant for the behaviours described).

Simplified models which abstract away a lot of the details are needed for theoretical understanding in neuroscience. However, arguments need to be made that the simplified model still maintains relevance for the system studied.

1. While the experimental results for connectivity included in the model are very detailed, they are not a complete description of the microcircuit. Most importantly, they lack a L4, which is the primary target of thalamic inputs, which is important if one studies studying evoked responses. They also lack a L6.

2. It is unclear how much of a cortical tissue would the model represent. If it is a small column, the authors need to justify the validity of studying it in isolation (rather than as a set of coupled microcircuits). If it corresponds to a significant fraction of an area, the authors need to justify the relevance of homogeneity assumption (that all neurons behave like the mean), or switch to methods representing heterogeneous populations.

3. While care was taken on the connectivity between individual populations, the input/output transformation is assumed identical. One would imagine that the effects studied of the circuit depend significantly on the intrinsic properties of the different cell types.

4. An unexpected choice for parameters for intrinsic time scales for the neurons involved which was only in the supplemental material on page 54: while the time scale for SST neurons is 30ms, for PV 7ms the time scale for the excitatory neurons is 3ms. I did try to follow the papers cited for this, but most are modeling papers, and in the experimental paper I did not find this justification. Much better referencing to the exact source of such data is needed. These choices are very different from the intrinsic membrane time constant for these cells.

[Editors’ note: further revisions were suggested prior to acceptance, as described below.]

Thank you for resubmitting your work entitled "Rate and Oscillatory Switching Dynamics of a Multilayer Visual Microcircuit Model" for further consideration by *eLife*. Your revised article has been evaluated by Floris de Lange (Senior Editor) and a Reviewing Editor.

The manuscript has been improved but there are some remaining issues identified by the first reviewer that need to be addressed, as outlined below:

*Reviewer #1 (Recommendations for the authors):*

The authors addressed reasonably 4 out of the 5 original points.

Points 1 (lack of layers 4/6) and 2 (homogeneity of inputs/connections) are addressed, and they point to limitations of the model. Layer 4 is discussed but only as an input (while biologically it is strongly recurrently connected) and Layer 6 seems not to be discussed. Given the experimental data available I believe nothing needs to be studied in the results, but a small Discussion section with limitations could help put the work in context.

However, the point about time scales remains incredibly confusing to me.

A critical point is the reply from the authors:

"To test the impact of such time constant differences, we systematically varied the synaptic time constants across different neuron types and examined the network oscillation frequency after VIP input (see the newly added Figures 5e-h, lines 355-375). The notable result was that the transition between a slow and high frequency oscillation remained clearly visible even when all neuron types had the same time constant. This supports the conclusion that the frequency switch is already engrained in the connectivity of the network and is only modified by differences in time constants."

Yet looking at figure 5h, for the same time constants (orange line) the transition is not clearly visible to me.

Another point which is somewhat confusing: looking at equation 1, $\tau$ seems to be better mapped to the membrane time constant and looking at equation 2 the synapses are instantaneous. Yet the authors point to choices of $\tau$ to synaptic not membrane time constants.

If I am interpreting correctly figure 5h, it seems that an important effect described disappears when the time constant for pyramidal neurons reaches 10ms, which is on the low end for the membrane time constant for excitatory neurons. A clear explanation why $\tau$ in equation 1 is mapped by the authors to synaptic rather than membrane time constant, and why the membrane time constant can be ignored when it is longer than the synaptic time constant is needed to understand the applicability of the model for biological circuits.

---

## [Author Response]

[Editors’ note: the authors resubmitted a revised version of the paper for consideration. What follows is the authors’ response to the first round of review.]

Reviewer #1:In this study, the authors build a microcircuit of mouse primary visual cortex using connectivity data between four cell types (PV, SSt, Pyr, VIP) in two laminar compartments (superficial and deep). The authors then examine the influence of (1) Global increase in coupling (G), and (2) Specific drive to particular neurons. This basic microcircuit appears to capture some basic features of physiology, namely: (1) Increased high-frequency power in superficial layers and increased low-frequency power in deep layers (Figure 1), (2) Enhanced firing rates in deep layers. The main finding of the authors is that when they enhance drive to the VIP cells, there is a steep transition in the firing of SST cells, which they characterize as a bi-stable system. Overall, this study has merit in providing a quite extensive characterization of a canonical microcircuit, and the authors do a great job in linking their manipulations to experimental findings.

We thank the reviewer for this important comment. Indeed, VIP neurons are inhibitory, but the net effect of activating these neurons is contingent on the activity of SST neurons. The reason a disinhibitory peak cannot be found with lower G-values is that the SST rates are low and VIP/PV cells are more active, such that further drive to these cells only leads to a small further suppression of SST cells. As a consequence, there is no release from SST inhibition and VIP (or PV) cells start to inhibit PYR cells immediately. By contrast, when SST neuron rates are sufficiently high, the initial suppression of SST cells releases PYR neurons from inhibition and leads to a disinhibitory peak before the VIP/PV inhibition starts to inhibit PYR neurons again. Spontaneous SST rates are enhanced by larger G-values and thus the disinhibition in Figure 3a and b are solely seen for high G-values. We also provided external SST drive and now show that with more input the disinhibition also increases (Supplementary Figure 3b, lines 249-253).

Concerns:1) The authors describe the effect of driving VIP interneurons as disinhibitory. I'm confused. Figure 3 shows that the effect of driving VIP interneurons in pyramidal firing rates in superficial layers is largely suppressive, which seems to go against their main conclusion and experimental findings.

The oscillations cannot be described as a limit cycle, since the model does not display oscillations in the absence of noise. The network is rather poised to a focus equilibrium and oscillations are thus due to resonance which are unmasked in the presence of noise only. We clarify this point now at the beginning of the Results section (see lines 118-120).

2) Noisy oscillations should be better defined. Are these noisy limit cycles or quasi oscillations?

We thank the reviewer for this suggestion. Switching dynamics that include transitions between oscillation frequencies in simplified networks have been found in studies in which a cell population was modeled by either several Wilson-Cowan units or spiking neuron models (Hertaeg et al., 2019, PlosCompBio, Lee et al., 2018, Cell Reports). More recently Kim et al. (2019 *PNAS*) have shown that for a wide range of parameters firing rate models can be mapped on networks with spiking neurons. Therefore, we expect that most of our results will also be observed in a network with spiking neurons. Indeed, our preliminary work has confirmed this [unpublished observations by *Guo and Kumar].* This is now discussed in the methods section ('population activity model') (lines 733-736).

3) The model provided by the authors has a stochastic drive and therefore exhibits noisy oscillations. It should be discussed whether similar results would have been obtained when using stochastic Wilson-Cowan models where each population consists of multiple neurons and the neurons have spiking output.

We thank the reviewer for this suggestion. Switching dynamics that include transitions between oscillation frequencies in simplified networks have been found in studies in which a cell population was modeled by either several Wilson-Cowan units or spiking neuron models (Hertaeg et al., 2019, PlosCompBio, Lee et al., 2018, Cell Reports). More recently Kim et al. (2019 *PNAS*) have shown that for a wide range of parameters firing rate models can be mapped on networks with spiking neurons. Therefore, we expect that most of our results will also be observed in a network with spiking neurons. Indeed, our preliminary work has confirmed this [unpublished observations by *Guo and Kumar].* This is now discussed in the methods section ('population activity model') (lines 733-736).

4) A limitation of the study is that Layer 4 is not modelled, and that the application of external drive appears somewhat arbitrary. In reality the external drive on PV and Pyramidal cells will likely be matched/balanced to some extent. However, Figure 5 applies a constant input to either PV or VIP cells which does not seem a realistic assumption. It would be hard to make reasonable assumptions on how the input drive to VIP cells looks like.

To study the microcircuit, we applied drive to single cell types analogous to optogenetic stimulation to compare with the experimental literature and make predictions. As pointed out correctly this is not what may be happening under in vivo conditions in the neocortex, where all cell types get simultaneous input from other circuits or within the circuit such as the superficial layer from layer 4 in V1. However, this simultaneous input is not uniform to all cell types and is scaled as a function of input type. In mouse V1, it was found that layer 4 pyramidal cells preferably drive PV and other PYR cells in the superficial layer, skipping VIP cells but have little influence on SST cells (Adesnik et al., 2012, Nature). In contrast, feedback drive form higher order areas to the V1 superficial layer of mice targets mainly VIP cells and to a much lesser extent all the other cell types (Zhang et al., 2014, Science).

However, the reviewer’s concern is valid and therefore we have now added a new figure where we show the effect of stimulating all cell types in the superficial layer together, scaling the input to the different cell types to mimic drive from L4 or feedback input from the anterior cingulate cortex (Figure 9 and lines 469-489). We found that as G increased the activity of SST cell decreased compared to the other cell types and subsequently the network switched from an inhibited, slow oscillation dominated state, to a disinhibited state with high frequency oscillations. This is consistent with reports that visual stimulation of V1 replace slow oscillations by γ oscillations (Chen et al., 2017, Veit et al., 2017).

5) Related to this, the authors identify that the bi-stable switch appears to emerge in a particular regime of G. However, do we have any idea what the value of "G" in an actual circuit would be? Furthermore, should one not examine how the strength of external drive modulates the ability of the circuit to switch?

The second reviewer also raised a question about the meaning of ‘G’. Therefore, we elaborate on the meaning of the G-value in more detail in the results (see lines 113-120) and Discussion sections (lines 635-647). Also, based on the experimental literature we speculate in the discussion what values the G parameter may assume in the mouse V1 (lines 639-643).

To address the second concern, we applied a constant drive to all populations in the circuit and systematically varied its strength (see newly added Supplementary Figure 7). We found that stronger baseline activity enhanced the disinhibition effect and shifted SST response curves to higher values, while the slope of the sigmoidal curves remained unchanged. Thus, external drive does not introduce qualitative changes in network behavior and features such as a transition from ultrasensitivity to bistability are observed for a large range of input values.

6) A very relevant study, Di Poppa et al. by Ken Harris lab, is only marginally discussed in this paper. However, that paper also provided a model of interactions between these cell classes and reports many findings on disinhibition and covariation between the cell types.

The paper of Di Poppa et al. is relevant to our study and we mention its experimental findings in relation to our results on lateral inhibition in the context of surround suppression ('Lateral inhibition switches circuit to the SST state' in the Results section). We now also cite it to discuss the potential G-values in mouse V1 in the new paragraph of the Discussion section (lines 639-643). However, it is important to note that in terms of modeling Di Poppa et al. fitted the connections to estimate effective interactions between cell populations under different experimental situations related to locomotion. By contrast, our study is based solely on anatomical connections and their weights as provided by the experimental literature (Jiang et al., 2015) and no fitting or adjustment of connections was performed, except for testing the causal role of specific links for microcircuit properties.

7) The authors should stress that this is a model of the MOUSE primary visual cortex. We know that the monkey visual cortex has quite different properties in terms of cell types, connectivity, lamination etc. It's unclear whether the mouse work bears any relevance for the monkey. The authors mix references to mice and monkey studies, but this should be made clear at all point. For instance, γ oscillations in monkeys have quite different signatures from the mouse in terms of frequencies. We can't really compare those in a straightforward way.

We have now added a new paragraph in the Discussion section (see lines 652-651), where we discuss experimental findings that are similar between mouse and monkey studies and which our model was also able to replicate, such as the oscillation difference between layers and the frequency drop in the presence of surround suppression.

8) The study relies on the octo-patch data. However, that dataset was criticized quite severely in a commentary. Can references be made to other studies e.g. from Scanziani confirming some of these connectivity patterns?

Anatomically, one of the main concerns of the Jiang et al. 2015 study pointed out by other experimentalists in a commentary was that recurrent connections between pyramidal cells were very weak or absent, which was related to the age of the animals. To test how recurrent excitation influences our results we parametrically changed the connection strength of the PYR cell self-connection. We curiously found that the switching dynamics of the circuit is fully preserved even in the absence of recurrent excitation (see lines 418-427). Moreover, more recurrent excitation only had a slight impact on the slope of sigmoidal response curves and reduced the frequency of the high-frequency oscillation during the disinhibited state.

Reviewer #2:This is an excellent, innovative, well written and comprehensive modeling study. The paper is made possible with detailed anatomical data that is only available recently about connectivity matrices and cell type distributions across layers. The study systematically shows how drive to VIP (feedback type) and PV (feedforward type) cell modulate a SST cell dependent inhibited and a PV dependent disinhibited state. The study mechanistically tracks down the source for the two separable states asymmetric self-inhibition and show how state transitions can be induced ('toggled'). The paper provides strong quantifiable predictions which is attractive for empirical people.A weakness is that many assumptions are implicit and key findings are difficult to find in the large amount of data.1. It is difficult to get a good understanding which G values are realistic for invite states and what they mean. There is a rate and power change for G of < 100 (Figure 1), but then various interesting effects are occurring at G >200 (effects of cell silencing).What are the underling characteristics / mechanisms that determine g changes, i.e. changes in effective coupling strength? describing this more explicitly early in the results would enhance comprehensibility

To address this question, we have now expanded the explanation of the G-value at the beginning of the Results section (see lines 111-117) and also added a paragraph in the discussion (lines 631-643). The reason why we introduced the scaling factor G is that the empirical connectivity data provided by Jiang et al. (2015) only describe PSP strength and connection probability between pairs of single cells. However, in our simulations we modeled the activity of the entire population of each neuron class (interneuron types and pyramidal cells) using a single differential equation which mimics the mean firing rate of the entire population of a single neuron types. These differential equations were then coupled according to the connectivity matrix derived from single cell connections. However, a population of neurons has a much stronger impact on another population within the microcircuit than the impact of a single PSP on a single neuron. As we don't know the absolute size of each population in the circuit, we introduced the parameter G that scales the connections between the neurons. As this parameter increases, the interaction strength between the neuronal populations increases, which is analogous to increasing the absolute size of the cell class populations, or the number of cells per class. In the discussion we also speculate based on empirical results where the G-value of the V1 mouse cortex may lie.

2. The presentation of the key contributions of SOM and PV is ideally improved. Going through Figure 2 is highly interesting but time consuming because the many panels make this a dense figure. Can the two key results of the effects of PV silencing and SOM silencing be shown as a power vs frequency plot – identical to Figure 2C and D. So far, we only see the peak frequencies across G but this does not easily allow to appreciate the power spectral shape at a low (e.g. 50) or high (e.g. 450) G. In its current form it is not possible to confirm how wide the power spectral density peaks are.

We appreciate this suggestion. However, replacing the summary figures of Figure 2, where the peak frequency is shown as a function of G-values, would increase the number of figures considerably. As there are already a number of additional main figures added after the revision, we prefer to keep the current layout of Figure 2, and hope for the reviewer’s understanding.

3. the results are describing average effects and are surprisingly non-quantitative. There is not a single statistic used and error bars are lacking. While this seems justified in many cases it leaves the impression that there is no noise in the results and that they would be perfectly reproducible and robust across may conditions. How realistic is this and what would change if realistic noise is introduced?When the authors address this question, it would be important to see an explicit definition of what a 'state' refers to. The paper describes many simulations where inputs are switched on constantly and with no apparent fluctuations. Is that realistically happening in real feedforward /feedbacK signal dynamics?

We thank the reviewer for raising this point. In order to address this concern, we have added Supplementary Figure 8a-c. First, we added Gaussian white noise to all the cells in the model and measured the response of SST neurons after VIP input, as the switching properties can be seen most clearly here. It can be observed that the mean of 50 noise trial does not differ from a single noise free simulation (see Supplementary Figure 8a, lines 317-320).

Next, we added a different type of noise to the model by jittering the connectivity weights, as the empirical data in Jiang et al. (2015) are presented as mean and standard deviation. Jittering caused changes in the slope of sigmoidal response curves, but this type of jitter did not change the ultrasensitivity properties of the circuit per se (see Supplementary Figure 8b,c, lines 321-327).

Showing all the results in the presence of noise will make the figures rather unwieldy. Therefore, we think that it is sufficient to show that the key results do not change in the presence of noise, while leaving the rest of the figures unchanged.

In our manuscript, a state refers to a different excitation-inhibition balance, which is characterized by the prevalent activity of a specific interneuron type (lines 85-89). Here, we externally drove either SST or PV cells, analogous to optogenetic drive from outside, and found that when PV or VIP cells are strongly driven there was more excitation in the network as compared to SST cell drive, which overall inhibits the circuit much more strongly. If the network is in a relatively more excited state, passing activity from this network to further downstream circuits is easier than in more inhibited states.

These simulations were adapted to show what happens in case of optogenetic activation. However, in a more realistic scenario, bottom-up and top-down drive will activate all cell types simultaneously as shown experimentally, albeit with different weights (see e.g. Adesnik et al., 2012 for layer 4 drive to the superficial layer in mouse V1, or see Zhang et al., 2014 for top-down drive from the anterior cingulate to V1 in the mouse). Thus, we modeled such scenarios by driving all cell types simultaneously in the superficial layers weighted by empirical data (Adesnik et al., 2012 for bottom-up drive and Zhang et al., 2014 for top down drive). We found that both types of drive tend to suppress SST activity as G-values increase and lead to a transition from slow oscillator activity to a high frequency oscillation (Figure 9, lines 469-489), in accordance with experimental findings that both bottom up ('visual stimulation') and top-down activation ('attention) suppress slow oscillations in favor of a high frequency γ rhythm.

Reviewer #3:Using a Wilson-Cowen model, the authors look at the coupling between 8 neuronal populations (excitatory + 3 inhibitory) based on experimental connectivity of the L2/3 and L5 populations. They do a very thorough analysis of the oscillatory properties of this system under a set of assumptions. Even though they do not address computations directly, the results are interesting, and link to several hypotheses in the field on the generation of oscillatory behavior. However, I am not convinced that following the choices of parameters and approximations the model is a useful approximation of the biological circuit. I believe the authors need to do considerably more work to justify the choices of parameters and the implicit approximations used (or that they are irrelevant for the behaviours described).Simplified models which abstract away a lot of the details are needed for theoretical understanding in neuroscience. However, arguments need to be made that the simplified model still maintains relevance for the system studied.1. While the experimental results for connectivity included in the model are very detailed, they are not a complete description of the microcircuit. Most importantly, they lack a L4, which is the primary target of thalamic inputs, which is important if one studies studying evoked responses. They also lack a L6.

The original connectivity matrix as reported by Jiang et al. (2015) indeed did not include layers 4 and 6. Even though the cell composition of layer 4 in mouse V1 has been studied in detail (Scala et al., 2019, Nat Comm), its connectivity with the other layers and their cell types is not known. To circumvent the problem and study the impact of layer 4 activation on the circuit, we modeled layer 4 input to the superficial layer based on a physiological study (Adesnik et al., 2012, Nature), which charted the EPSP strength in different cell types of the superficial layer after stimulation of L4. In our model, we thus applied excitatory input to all superficial neuron types that was scaled by weights provided by the empirical EPSP strength. We used a similar approach to model feedback input to the superficial layer of mouse V1 on the basis of similar experimental results obtained by Zhang et al. 2014 (Science), which reported EPSP strength of different cell types in V1 after stimulation of the anterior cingulate of the mouse. The results are shown in newly added Figure 9 (see also lines 469-489) showing that both bottom-up drive (visual stimulation) or top-down stimulation (e.g. attention) can switch the network from a low to a high frequency oscillation state.

2. It is unclear how much of a cortical tissue would the model represent. If it is a small column, the authors need to justify the validity of studying it in isolation (rather than as a set of coupled microcircuits). If it corresponds to a significant fraction of an area, the authors need to justify the relevance of homogeneity assumption (that all neurons behave like the mean), or switch to methods representing heterogeneous populations.

In the original paper by Jiang et al. (2015), the connectivity matrix was constructed for interneurons that were <250 micrometers apart within a layer, as connectivity quickly dropped beyond this distance. Thus, our microcircuit model also covers a similar spatial extent in the horizontal plane. Note that the influence of neighbouring circuit was studied in our manuscript by driving SST cells externally, mimicking lateral inhibition mediated by horizontal connections in mouse V1 (see lines 384-396).

The connectivity matrix in Jiang et al. (2015) contained both mean values for the connectivity strength (PSP strength and connection probability) as well as their standard deviation. We used the mean connectivity throughout the paper, but now we have also added a figure (Supplementary Figures 8 b-c, lines 321-327) in which the weights were jittered. The results show that the ultrasensitivity is preserved, while the slope of the sigmoidal response curve in SST cells varies with the jitter.

3. While care was taken on the connectivity between individual populations, the input/output transformation is assumed identical. One would imagine that the effects studied of the circuit depend significantly on the intrinsic properties of the different cell types.

To address this concern, we added two parameters to the transfer function and varied their values systematically. The results are shown in Supplementary Figures 8d-e and demonstrate that one parameter does not qualitatively change the results, only shifting sigmoidal curves to the right, while the other parameter has an influence on whether the network shows an ultrasensitive response or bistability. This is discussed in the revised manuscript (see lines 328-335)

4. An unexpected choice for parameters for intrinsic time scales for the neurons involved which was only in the supplemental material on page 54: while the time scale for SST neurons is 30ms, for PV 7ms the time scale for the excitatory neurons is 3ms. I did try to follow the papers cited for this, but most are modeling papers, and in the experimental paper I did not find this justification. Much better referencing to the exact source of such data is needed. These choices are very different from the intrinsic membrane time constant for these cells.

We thank the reviewer for this very helpful comment. We are happy report that further analysis of this issue yielded surprising results. Indeed, in the modeling literature the existence of slow and fast oscillations is often assumed and mechanistically implemented as a difference of synaptic time constants of different inhibitory populations (see e.g. Hyafil et al., 2015, Mejias et al., 2016). To test the impact of such time constant differences, we systematically varied the synaptic time constants across different neuron types and examined the network oscillation frequency after VIP input (see the newly added Figures 5e-h, lines 355-375). The notable result was that the transition between a slow and high frequency oscillation remained clearly visible even when all neuron types had the same time constant. This supports the conclusion that the frequency switch is already engrained in the connectivity of the network, and is only modified by differences in time constants.

[Editors’ note: what follows is the authors’ response to the second round of review.]

The manuscript has been improved but there are some remaining issues identified by the first reviewer that need to be addressed, as outlined below:Reviewer #1 (Recommendations for the authors):The authors addressed reasonably 4 out of the 5 original points.Points 1 (lack of layers 4/6) and 2 (homogeneity of inputs/connections) are addressed, and they point to limitations of the model. Layer 4 is discussed but only as an input (while biologically it is strongly recurrently connected) and Layer 6 seems not to be discussed. Given the experimental data available I believe nothing needs to be studied in the results, but a small Discussion section with limitations could help put the work in context.

We thank the reviewer for this additional comment. We now added a section about anatomical limitations of our study to the discussion, mentioning that there is a lack of detailed information about the connectivity within layers 4/6. Moreover, studies about their cell-specific connections to the other layers are missing.

However, the point about time scales remains incredibly confusing to me.A critical point is the reply from the authors:"To test the impact of such time constant differences, we systematically varied the synaptic time constants across different neuron types and examined the network oscillation frequency after VIP input (see the newly added Figures 5e-h, lines 355-375). The notable result was that the transition between a slow and high frequency oscillation remained clearly visible even when all neuron types had the same time constant. This supports the conclusion that the frequency switch is already engrained in the connectivity of the network and is only modified by differences in time constants."Yet looking at figure 5h, for the same time constants (orange line) the transition is not clearly visible to me.Another point which is somewhat confusing: looking at equation 1, $\tau$ seems to be better mapped to the membrane time constant and looking at equation 2 the synapses are instantaneous. Yet the authors point to choices of $\tau$ to synaptic not membrane time constants.If I am interpreting correctly figure 5h, it seems that an important effect described disappears when the time constant for pyramidal neurons reaches 10ms, which is on the low end for the membrane time constant for excitatory neurons. A clear explanation why $\tau$ in equation 1 is mapped by the authors to synaptic rather than membrane time constant, and why the membrane time constant can be ignored when it is longer than the synaptic time constant is needed to understand the applicability of the model for biological circuits.

To address this concern, we have recalculated Figure 5h to better depict the frequency changes with increase in VIP input. The figure shows now more clearly that while the disinhibition peak with enhanced frequency decreases with higher PYR time constants, it remains at approximately double the size of the SST dominated baseline activity before the disinhibition starts (see Results section, oscillations and switching dynamics section). The slower time constants of PYR cells also slows down all inhibitory cell types which have a joint effect on decreasing the peak (especially SST and PV cell time constants influence the peak, see Figure 5e-f).

We also followed the reviewer's advice and explained the meaning of time constants in the Wilson Cowan model in more detail in the methods section (population activity model section). In the Wilson-Cowan model, the time constant refers to the population of neurons and it is not possible to associate it to either synaptic or membrane time constants. Therefore, we have deleted all the references to 'synaptic' time constants in the manuscript.